# NopeRoomGS: Indoor 3D Gaussian Splatting Optimization without Camera Pose Input

**Wenbo Li**[1,2*]    **Yan Xu**[1,3* ✉]    **Mingde Yao**[1,2]    **Fengjie Liang**[4]    **Jiankai Sun**[5]
**Menglu Wang**[6]    **Guofeng Zhang**[7]    **Linjiang Huang**[8]    **Hongsheng Li**[1,2]

[1]MMLab, The Chinese University of Hong Kong    [2]CPII    [3]University of Michigan
[4]Hong Kong Polytechnic University    [5]Stanford    [6]USTC    [7]Zhejiang University    [8]BUAA
wenboli@zju.edu.cn, yxumich@umich.edu,
zhangguofeng@zju.edu.cn, ljhuang@buaa.edu.cn, hsli@ee.cuhk.edu.hk

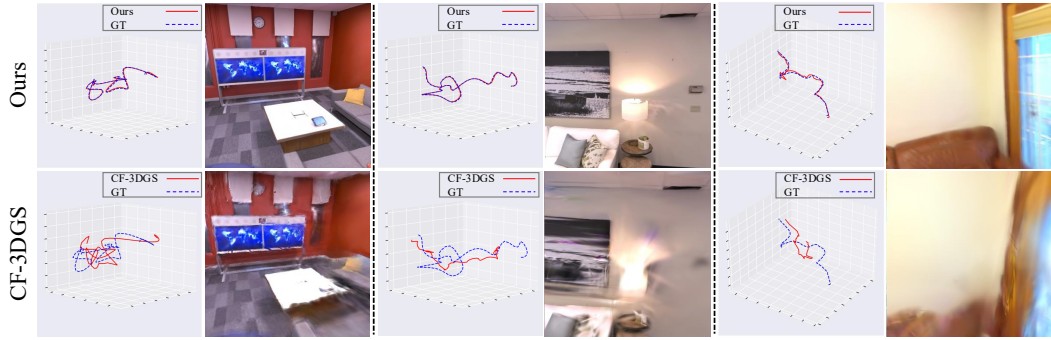

Figure 1: Comparison of camera pose estimation and novel view synthesis with the state-of-the-art. Compared to CF-3DGS [10] (bottom), our method (top) achieves more robust pose estimation and more photorealistic novel view synthesis in challenging indoor scenes with textureless regions or abrupt camera motion. Each example contains the camera trajectory estimation (left) and a sampled synthesized view (right).

## Abstract

Recent advances in 3D Gaussian Splatting (3DGS) have enabled real-time, high-fidelity view synthesis, but remain critically dependent on camera poses estimated by Structure-from-Motion (SfM), which is notoriously unreliable in textureless indoor environments. To eliminate this dependency, recent pose-free variants have been proposed, yet they often fail under abrupt camera motion due to unstable initialization and purely photometric objectives. In this work, we introduce **Nope-RoomGS**, an optimization framework with **no** need for camera **po**se inputs, which effectively addresses the textureless regions and abrupt camera motion in indoor **room** environments through a local-to-global optimization paradigm for 3D**GS** reconstruction. In the local stage, we propose a lightweight local neural geometric representation to bootstrap a set of reliable local 3D Gaussians for separated short video clips, regularized by multi-frame tracking constraints and foundation model depth priors. This enables reliable initialization even in textureless regions or under abrupt camera motions. In the global stage, we fuse local 3D Gaussians into a unified 3DGS representation through an alternating optimization strategy that jointly refines camera poses and Gaussian parameters, effectively mitigating gradient interference between them. Furthermore, we decompose camera pose optimization based on a piecewise planarity assumption, further enhancing robustness under abrupt camera motion. Extensive experiments on Replica, ScanNet and

---

* Co-first authorship.
✉ Corresponding author.

39th Conference on Neural Information Processing Systems (NeurIPS 2025).

Tanks & Temples demonstrate the state-of-the-art performance of our method in both camera pose estimation and novel view synthesis.

# 1 Introduction

The reconstruction of 3D representations from images for photorealistic novel view synthesis is a longstanding challenge in computer vision and graphics. It has broad applications in augmented reality, virtual reality, and robotics. Recent advances in learning-based methods [35, 2, 49, 51, 25, 31] have significantly improved rendering fidelity and generalization. However, the majority of these approaches still rely on externally estimated camera poses, typically obtained via Structure-from-Motion (SfM) pipelines [35, 49]. This dependency introduces a critical vulnerability: SfM is notoriously unreliable in low-texture regions and under non-smooth camera motion [12, 13, 32], where sparse or ambiguous feature matches lead to pose estimation failures. Such inaccuracies corrupt reconstruction optimization, but also degrade synthesis quality [38, 18, 42, 7], entangling final novel view synthesis quality with the success or failure of an external, heuristic-heavy preprocessing step.

This misalignment has motivated efforts to remove SfM entirely and jointly optimize pose and scene representation [5, 31, 53, 62, 20, 10]. However, this joint optimization problem presents a classic chicken-and-egg dilemma, requiring careful algorithmic design. BARF [31] addresses this by employing a progressive frequency scheduling strategy, gradually increasing the Fourier components in NeRF optimization to stabilize camera pose estimation while preserving accurate scene reconstruction. Nope-NeRF [5] introduces inter-frame geometric constraints and enhances the pose representation to improve stability. More recently, 3D Gaussian Splatting (3DGS) [25] has emerged as a powerful alternative to NeRF, offering both higher rendering efficiency and superior visual quality. Building on this representation, Fu et al. [10] propose the first framework for pose-free novel view synthesis. To facilitate convergence, their method incrementally grows 3D Gaussians frame by frame as the camera moves, optimizing each frame's pose solely through photometric error minimization.

Despite significant advancements, existing pose-free methods often assume that the photometric error from visual textures alone can provide strong and reliable gradients for accurate pose optimization. However, this assumption frequently breaks down in the presence of textureless regions or abrupt camera, e.g., in indoor

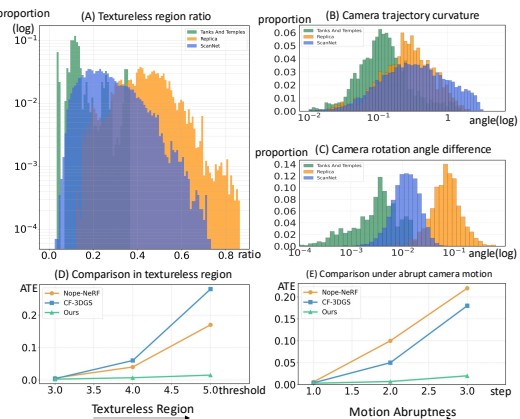

Figure 2: The distributions of low-texture regions (A), camera trajectory curvature (B), and camera rotation difference (C) vary significantly across different scenarios: Tanks & Temples [27], Replica [44], and ScanNet [9]. The indoor datasets, Replica and ScanNet, contain more textureless regions and exhibit more abrupt camera motions than Tanks & Temples, which is generally texture-rich and captured under more controlled conditions. To create more challenging scenarios, we further mask textured regions and sample frames with larger temporal steps on Tanks & Temples. Under these settings, our method demonstrates strong robustness to large textureless areas and sudden camera motions, while Nope-NeRF [5] and CF-3DGS [10] show notable performance drops (D, E).

environments. We evaluated several state-of-the-art methods on indoor datasets and observed notable performance degradation (see Tab. 1 for details). Furthermore, when we artificially mask texture-rich regions or increase the camera motion speed within the datasets originally used by these methods, we observe similarly pronounced performance drops (Fig. 2). These findings empirically validate our hypothesis regarding the limitations of the current approaches in indoor environments.

To overcome these limitations, we propose NopeRoomGS, a robust pose-free 3DGS framework designed for scenarios where textureless regions and abrupt camera motion prevail, especially indoor environments. Our method employs a local-to-global scheme to progressively build up the 3DGS.

In the local stage, we introduce a lightweight local neural geometric representation that jointly optimizes depth and camera pose on short, overlapping video clips extracted from the input sequence. To recover the consistent geometry of each video clip, the optimization is supervised by multi-frame tracking constraints derived from CoTracker [23], which remain effective even under abrupt camera

motion. Furthermore, to enhance robustness in textureless regions, we incorporate regularization from a pretrained monocular depth foundation model [24]. This design enables reliable recovery of local 3D geometry and camera poses under challenging conditions with textureless regions and abrupt camera motion, thereby providing a stable initialization of local 3D Gaussians for subsequent global fusion.

In the global stage, we progressively fuse these local 3D Gaussians into a unified global 3DGS representation over entire video sequence. This fusion process is formulated as an optimization problem, where the global 3DGS is supervised by a combination of photometric loss, depth alignment loss, and pose constraints derived from piecewise planarity assumption, ensuring global geometric consistency. Furthermore, we adopt an alternating optimization strategy that updates Gaussian parameters and camera poses in turn, rather than jointly, to reduce gradient interference and enhance convergence stability. This strategy improves the robustness of global reconstruction in challenging scenarios with textureless regions and abrupt camera motion.

Our contributions are summarized as follows:

- We propose a pose-free 3DGS framework with a local-to-global scheme, which exhibits strong robustness in textureless regions and under abrupt camera motion.

- In the local stage, we introduce a lightweight neural geometric representation that jointly optimizes depth and camera pose over short video clips. It yields locally consistent geometry and accurate poses, even in textureless regions and under abrupt camera motion, effectively bootstrapping a set of local 3D Gaussians.

- In the global stage, we progressively fuse local 3D Gaussians into a unified global 3DGS representation with alternative optimization strategy. To ensure stable convergence, we supervise the model with the combination of photometric loss, depth alignment loss, and pose constraints derived from piecewise planarity assumption. These components collectively enforce structural consistency in the global reconstruction.

- Experiments on public datasets, including Replica [44] and ScanNet [9] and Tanks & Temples [27], demonstrate that our method achieves state-of-the-art performance, with particularly strong results in indoor scenes and competitive results in general scenarios.

## 2  Related Work

**Novel view synthesis**. Generating photorealistic images from novel viewpoints is a key challenge in computer vision, which has been approached using diverse 3D representations. Among them, Neural Radiance Fields (NeRFs) [35] have emerged as a leading method, with extensions tackling challenges such as aliasing [2, 3, 4], surface reflectance [48, 1], sparse views [26, 37, 55, 60, 19]. In parallel, explicit point-based or mesh-based representations have gained increasing attention for their efficiency and interpretability [56, 65, 25, 34, 28, 64, 33]. In particular, 3DGS [25] demonstrates that real-time, high-fidelity view synthesis can be achieved by representing scenes as sets of anisotropic Gaussian primitives optimized through differentiable rendering. However, many of these methods still rely on pre-computed camera poses. Although the camera pose estimation has been studied for decades [36, 43, 58, 57, 47, 16, 17, 50], it is still challenging to robustly estimate the camera poses with low-cost sensors. Most recent novel view synthesis methods rely on Structure-from-Motion tools like COLMAP [11, 43, 36, 46, 33], which limit their applicability in scenarios where obtaining accurate camera poses is challenging, such as in low-texture environments or when only sparse or unstructured image collections are available.

**Radiance field without camera pose prior**. Recent studies have integrated pose estimation into NeRF training to remove dependence on pre-computed camera poses. Early methods, such as i-NeRF [63], refined camera parameters by aligning keypoints to a pre-trained NeRF, while NeRFmm [54] jointly optimized both the NeRF model and camera poses, albeit with limitations to forward-facing scenes. BARF [31] addressed gradient inconsistency in positional encoding with coarse-to-fine optimization, but still required initial pose estimates within 15° of the ground truth.

Nope-NeRF [5] leveraged monocular depth priors for both scene reconstruction and pose estimation. Despite these advances, NeRF-based methods remain computationally intensive and face challenges in achieving real-time rendering performance [40].

**3DGS without camera pose prior.** Recent work such as CF-3DGS [10] first removes COLMAP dependency by jointly optimizing poses and Gaussians under inter-frame photometric supervision, making them susceptible to failure under large camera motion or textureless regions. In parallel, a growing line of feed-forward, pose-free 3DGS methods [61, 22, 15, 8] propose generalizable

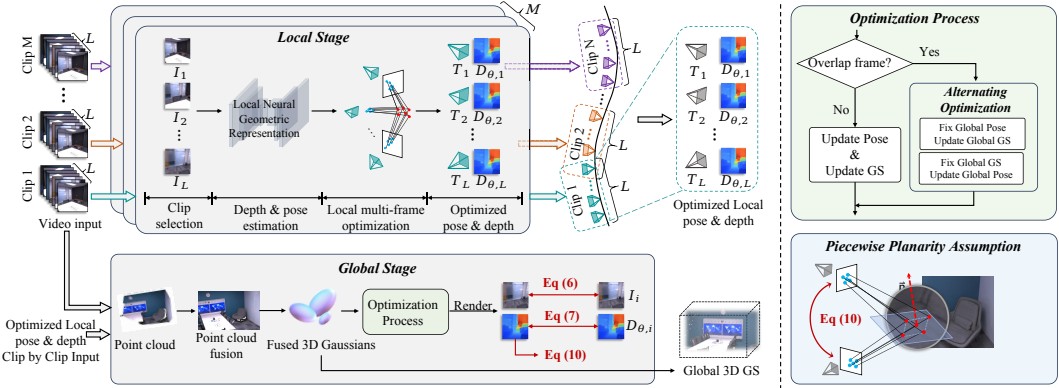

Figure 3: The pipeline of NopeRoomGS. Our NopeRoomGS framework adopts a Local-to-Global optimization scheme to address textureless regions and abrupt camera motion. In the local stage (Sec. 3.1), a lightweight Local Neural Geometric Representation jointly optimizes depth and camera pose for each clip, yielding consistent geometry and local 3D Gaussians. In the global stage (Sec. 3.2), these are fused into a unified 3DGS via differentiable optimization with photometric, depth, and pose constraints under the Piecewise Planarity Assumption. An Alternating Optimization strategy mitigates gradient interference, ensuring stable and accurate reconstruction.

networks to predict Gaussians (and often poses) without per-scene optimization. NoPoSplat [61] predicts 3D Gaussians directly in a canonical space, whereas SelfSplat [22] jointly learns camera poses and scene geometry via self-supervised depth and photometric consistency. Although these approaches remove the need for external SfM, they still face limitations when processing very large image collections, where computational and memory costs grow sharply and maintaining stable global pose/geometry estimates becomes challenging.

In contrast, we fully exploit 3D structural-consistency constraints and foundation-model priors with a lightweight local neural geometric representation, and fuse the resulting local 3D Gaussians using alternating optimization strategy of camera poses and 3D Gaussian parameters, which stabilizes initialization, reduces gradient interference, and yields globally consistent reconstructions in challenging scenarios with textureless regions and abrupt camera motion.

## 3 Method

Given a sequence of $N$ unposed images $\mathcal{I} = \left\{ I_i \in \mathbb{R}^{H \times W \times 3} \right\}_{i=1}^{N}$ with known camera intrinsic, our goal is to recover the camera poses $\mathcal{T} = \left\{ \mathbf{T}_i \in \mathbb{SE}(3) \right\}_{i=1}^{N}$ and reconstruct a unified global 3DGS representation $\mathcal{G}$ of the scene.

This task is fundamentally challenging because both accurate camera poses and reliable 3DGS representation must be recovered simultaneously from unposed inputs. The explicit nature of the 3DGS representation, though beneficial for efficient rendering, makes this joint optimization highly non-convex and sensitive to initialization [21, 6]. In particular, the problem is exacerbated in indoor environments characterized by textureless surfaces and abrupt camera motion, where standard feature-based methods fail and gradient descent is prone to local minima [14]. To improve robustness, we adopt a local-to-global optimization scheme, as illustrated in Fig. 3. In the local stage (Sec. 3.1), we introduce a learnable local neural geometric representation that jointly estimates depth and camera pose within each video clip. This yields reliable scene structure and camera pose to bootstrap a set of local 3D Gaussians as an initialization for global optimization, which is critical for stable convergence [59]. In the global stage (Sec. 3.2), these local 3D Gaussians are progressively fused into a unified global 3DGS representation. This fusion is formulated as a differentiable optimization problem, supervised by a combination of photometric loss, depth alignment loss, and pose constraints based on a piecewise planarity assumption. In addition, we adopt an alternating optimization strategy to updates camera poses and Gaussian parameters in turn, rather than jointly [30], to mitigate gradient interference and enhance convergence.

### 3.1 Local Stage

In the local stage, to address the challenging scenario with textureless regions and abrupt camera motion, we propose a local neural geometric representation that jointly optimizes camera poses

and per-frame depth via gradient descent. The optimization is supervised by multi-frame tracking constraints [23], which enhance geometric coherence under abrupt camera motion. Additionally, we incorporate strong depth priors from a pretrained foundation model [24] to regularize the solution in textureless regions. Unlike prior pairwise methods [10], our representation is shared across all frames within a clip, enabling consistent geometric reasoning across different viewpoints.

**Local neural geometric representation.** To ensure consistent geometry under challenging conditions such as textureless regions and abrupt camera motion, we parameterize the local scene structure using a lightweight local neural geometric representation $\mathcal{F}_\theta$, where $\theta$ denotes learnable parameters. To make this representation both robust and rapidly deployable, $\mathcal{F}_\theta$ is initialized from a fast monocular depth estimator [41] and shared across all frames within a short video clip, enabling coherent geometry over the video clip.

Given a short clip $\mathcal{I} = \{I_i \in \mathbb{R}^{H \times W \times 3}\}_{i=1}^{L}$ of length $L$, the local neural geometric representation $\mathcal{F}_\theta$ takes as input a color image $I_i$ to predict the corresponding depth map $D_{\theta,i}$:

$$D_{\theta,i} = \mathcal{F}_\theta(I_i). \tag{1}$$

We denote the depth maps of the clip as $\mathcal{D}_\theta = \{D_{\theta,i} \in \mathbb{R}^{H \times W}\}_{i=1}^{L}$. By sharing the network $\mathcal{F}_\theta$ across the local video clip, consistent geometric cues can be learned during optimization. The intuition of this optimization process is similar to the classic bundle adjustment, but our approach is based on neural representations.

**Joint optimization of local camera pose and depth.** After obtaining the initial depth maps above, we jointly optimize the local camera poses $\mathcal{T} = \{\mathbf{T}_i \in \text{SE}(3)\}_{i=1}^{L}$ and the depth maps $\mathcal{D}_\theta$ by enforcing geometric consistency across frames.

Specifically, for any pixel $\mathbf{u}_i$ in the frame $i$, we can get its correspondences $\{\mathbf{u}_j | j \in \mathcal{N}(\mathbf{u}_i)\}$ in the neighboring frames $\mathcal{N}(\mathbf{u}_i)$ by using the quasi-dense off-the-shelf multi-frame tracking constraints derived from CoTracker [23]. To enforce the geometric constraints for pose optimization, we first unproject the pixel $\mathbf{u}_i$ to get its 3D position $\mathbf{p}_i$ in the current frame:

$$\mathbf{p}_i = \pi^{-1}(\mathbf{u}_i) = D_{\theta,i}(\mathbf{u}_i)\mathbf{K}^{-1}\begin{pmatrix} \mathbf{u}_i \\ 1 \end{pmatrix}. \tag{2}$$

where $\mathbf{K}$ denotes the intrinsic matrix and $D_{\theta,i}(\mathbf{u}_i)$ is its estimated depth value in Eq. 1. Then, we map it to the nearby frames and enforce the geometric consistency by minimizing the projection error:

$$\mathcal{L}_{\text{proj}}^i = \sum_{j \in \mathcal{N}(\mathbf{u}_i)} \left\| \pi\left(\mathbf{T}_{j \leftarrow i}\pi^{-1}(\mathbf{u}_i)\right) - \mathbf{u}_j \right\|^2, \tag{3}$$

where $\pi(\cdot)$ denotes the pinhole camera projection function given the intrinsic matrix $\mathbf{K}$, and $\mathbf{T}_{j \leftarrow i} \in \text{SE}(3)$ denotes the relative pose transformation from frame $i$ to frame $j$.

To avoid the depth optimization deviate too far away from the realistic solution manifold, we regularize its prediction with the output from a stronger but much heavier monocular depth foundation model [24], striking a balance between performance and efficiency. This regularization is implemented by a scale-and-shift-invariant regularization loss [41]:

$$\mathcal{L}_{\text{ssi}}^i = \rho\left(D_{\theta,i} - \alpha\tilde{D}_i - \beta\right), \tag{4}$$

where $D_{\theta,i}$ is the output from local neural geometric representation $\mathcal{F}_\theta$, and $\tilde{D}_i$ is the pseudo ground-truth produced by the foundation model, and $\rho(\cdot)$ is a distance function (i.e., Huber loss). The parameters $\alpha$ and $\beta$ are obtained by solving a least-squares problem [41] to resolve the scale ambiguity of monocular depth estimation.

The overall objective function in the local stage is thus:

$$\mathcal{T}, \mathcal{D}_\theta = \arg\min_{\mathcal{T}, \mathcal{D}_\theta} \mathcal{L}_{proj} + w\mathcal{L}_{ssi}, \tag{5}$$

where $\mathcal{L}_{proj}$ and $\mathcal{L}_{ssi}$ are the summation of all the $\mathcal{L}_{\text{proj}}^i$ and $\mathcal{L}_{\text{ssi}}^i$ respectively within the same clip, and the weight $w$ controls the regularization strength. Here, we slightly abuse the notation $\mathcal{D}_\theta$, as we optimize the parameter $\theta$ practically.

| BARF | NoPe-NeRF | CF-3DGS | Ours | GT |
|------|-----------|---------|------|-----|

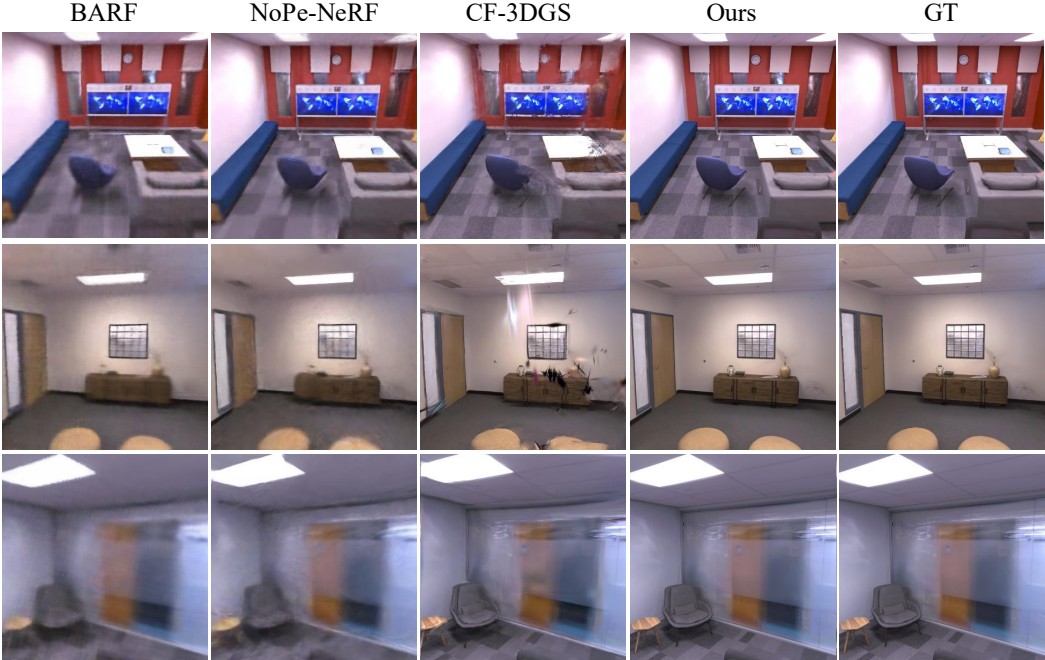

Figure 4: Qualitative comparison of novel view synthesis on Replica [44] dataset. Compared with other cutting-edge counterparts, our method synthesizes more detailed textures under the pose-free setting.

## 3.2 Global Stage

After obtaining the local camera poses $\mathcal{T}$ and depth maps $\mathcal{D}_\theta$ within each clip via Eq. 5, we initialize local 3D Gaussians accordingly and progressively merge them into a unified global 3DGS representation $\mathcal{G}$.

Specifically, for each frame $i$ within a clip, we unproject its depth map $D_{\theta,i}$ to the 3D space using camera intrinsic $\mathbf{K}$ (Eq. 2) to get the point cloud $\mathbf{P}_i$ and transform it using the estimated pose $\mathbf{T}_i$ to the local frame of this clip. After finishing this process, we get a local point cloud for each clip. To decrease the memory burden, we downsample the point cloud and initialize the local 3D Gaussians with the downsampled point cloud. Then, the local 3D Gaussians are merged into a unified global 3DGS representation $\mathcal{G}$ according to the relative pose between the local frame and the global frame. Thereafter, we optimize the global 3D Gaussian parameters as well as the camera pose based on the input image $\mathcal{I}$ and estimated depth map $\mathcal{D}_\theta$ from the local stage. In the ensuing part, we will elaborate on the objective function we used for optimization.

### 3.2.1 Objective Function

**Photometric loss.** Following the original 3DGS [25], we include the photometric loss terms in the objective function:

$$\mathcal{L}_{\text{rgb}}^i = \gamma \left\| I_{\mathcal{G}}(\mathbf{T}_i) - I_i \right\|_1 + (1 - \gamma)\mathcal{L}_{D-SSIM}(I_{\mathcal{G}}(\mathbf{T}_i), I_i), \tag{6}$$

which is constituted by a $L_1$ term and a D-SSIM term [25]. $I_{\mathcal{G}}(\mathbf{T}_i)$ represents the rendered image from the Gaussian Splatting $\mathcal{G}$ with the camera pose $\mathbf{T}_i$, $I_i$ is the $i$-th input image, and $\gamma$ is the hyperparameter to balance the two terms.

**Depth alignment loss.** With the photometric loss, the camera poses can be effectively corrected by the regions with rich textures, as these regions can produce strong gradients if the poses are erroneous. However, the indoor scenes are full of textureless areas, e.g., blank walls, which pose great challenges to pose-free 3DGS. To enhance the robustness in textureless regions, we also add a depth alignment term similar to Eq. 4:

$$\mathcal{L}_{\text{depth}}^i = \rho\Big( D_{\mathcal{G}}(\mathbf{T}_i) - \alpha \, D_{\theta,i} - \beta \Big). \tag{7}$$

The difference is that we here use the optimized depth maps $D_{\theta,i}$ from the local stage as the ground truth to constrain the depth maps $D_{\mathcal{G}}(\mathbf{T}_i)$ rendered with camera pose $\mathbf{T}_i$.

Table 1: Camera pose estimation performance metrics comparison on Replica [44] dataset. All baseline methods are trained with their official implementations and original configurations, and evaluated following the same protocol to ensure fair and consistent comparison. The best results are highlighted in bold.

| Scenes | Ours | | | CF-3DGS [10] | | | NoPoSplat [61] | | | SelfSplat [22] | | | Nope-NeRF [5] | | | BARF [2] | | | NeRFmm [54] | | |
|---|---|---|---|---|---|---|---|---|---|---|---|---|---|---|---|---|---|---|---|---|---|
| | $RPE_t\downarrow$ | $RPE_r\downarrow$ | $ATE\downarrow$ | $RPE_t$ | $RPE_r$ | ATE | $RPE_t$ | $RPE_r$ | ATE | $RPE_t$ | $RPE_r$ | ATE | $RPE_t$ | $RPE_r$ | ATE | $RPE_t$ | $RPE_r$ | ATE | $RPE_t$ | $RPE_r$ | ATE |
| office0 | **0.231** | **0.153** | **0.011** | 1.168 | 7.282 | 0.043 | 1.456 | 7.456 | 0.053 | 1.073 | 7.654 | 0.074 | 1.231 | 5.462 | 0.042 | 1.561 | 6.273 | 0.052 | 1.467 | 6.834 | 0.060 |
| office1 | **0.020** | **0.045** | **0.001** | 1.171 | 6.288 | 0.078 | 1.325 | 5.946 | 0.067 | 0.946 | 8.678 | 0.064 | 1.416 | 7.922 | 0.056 | 1.462 | 5.234 | 0.048 | 1.432 | 7.236 | 0.054 |
| office2 | **0.404** | **0.260** | **0.027** | 0.816 | 1.668 | 0.044 | 1.874 | 7.764 | 0.053 | 1.554 | 6.832 | 0.074 | 1.273 | 5.234 | 0.047 | 1.073 | 6.235 | 0.055 | 1.346 | 6.892 | 0.023 |
| office3 | **0.056** | **0.062** | **0.003** | 1.489 | 5.994 | 0.093 | 2.113 | 7.235 | 0.067 | 2.038 | 8.245 | 0.056 | 1.156 | 6.002 | 0.052 | 1.119 | 7.345 | 0.056 | 1.489 | 7.231 | 0.054 |
| office4 | **0.083** | **0.126** | **0.003** | 1.086 | 6.901 | 0.048 | 1.932 | 7.567 | 0.072 | 1.807 | 7.113 | 0.054 | 0.923 | 6.231 | 0.037 | 1.223 | 5.987 | 0.041 | 1.946 | 6.923 | 0.045 |
| room0 | **0.051** | **0.055** | **0.003** | 1.201 | 6.545 | 0.044 | 1.834 | 6.593 | 0.046 | 1.504 | 7.832 | 0.074 | 1.023 | 5.432 | 0.013 | 1.327 | 6.432 | 0.052 | 1.835 | 7.235 | 0.033 |
| room1 | **0.230** | **0.175** | **0.012** | 1.088 | 6.961 | 0.056 | 2.325 | 5.745 | 0.043 | 2.164 | 6.883 | 0.095 | 0.875 | 6.256 | 0.025 | 1.045 | 5.436 | 0.043 | 1.322 | 6.923 | 0.072 |
| room2 | **0.111** | **0.128** | **0.008** | 1.177 | 5.584 | 0.064 | 1.325 | 8.385 | 0.084 | 1.164 | 7.224 | 0.083 | 1.231 | 5.467 | 0.036 | 1.032 | 5.467 | 0.023 | 1.347 | 6.342 | 0.043 |
| mean | **0.148** | **0.126** | **0.009** | 1.150 | 5.903 | 0.059 | 1.773 | 7.086 | 0.061 | 1.531 | 7.56 | 0.072 | 1.141 | 6.001 | 0.039 | 1.230 | 6.051 | 0.046 | 1.523 | 6.962 | 0.048 |

**Pose constraint based on piecewise planarity assumption.** The camera motion could be abrupt in indoor scenes. To further improve the robustness against abrupt camera motion, we propose a cross-frame geometric constraint based on a piecewise planarity assumption.

We assume each point $\mathbf{x}_p$ in the scene lies on a infinitesimal piecewise plane defined by $\mathbf{n}_i^\top \mathbf{x}_p + \delta_i = 0$, where $\mathbf{n}_i$ is the surface normal and $\delta_i$ is the displacement coefficient. The normal and the displacement coefficient are calculated from the rendered depth map of 3DGS $\mathcal{G}$, using 4 surrounding pixels (left, right, upper, lower).

Given two consecutive frames, $I_i$ and $I_{i+1}$ with a relative pose estimation $\mathbf{T}_{i\rightarrow i+1} = [\mathbf{R}_{i\rightarrow i+1} \mid \mathbf{t}_{i\rightarrow i+1}]$ between, the plane parameters in frame $i$, denoted as $(\mathbf{n}_i, \delta_i)$, can be transformed to frame $i+1$ by

$$\hat{\mathbf{n}}_{i+1} = \mathbf{R}_{i\rightarrow i+1}\mathbf{n}_i, \tag{8}$$

$$\hat{\delta}_{i+1} = \delta_t - \mathbf{n}_i^\top \mathbf{t}_{i\rightarrow i+1}, \tag{9}$$

to get the transformed plane parameters $(\hat{\mathbf{n}}_{i+1}, \hat{\delta}_{i+1})$.

We then enforce consistency between the transformed plane parameters $(\hat{\mathbf{n}}_{i+1}, \hat{\delta}_{i+1})$ and the directly estimated values $(\mathbf{n}_{i+1}, \delta_{i+1})$ in the frame $i+1$, using the following loss function:

$$\mathcal{L}_{\text{plane}}^{i+1} = \lambda_n \left\| 1 - \hat{\mathbf{n}}_{i+1}^\top \mathbf{n}_{i+1} \right\|_2^2 + \lambda_\delta \left\| \hat{\delta}_{i+1} - \delta_{i+1} \right\|_2^2, \tag{10}$$

where $\lambda_n$ and $\lambda_\delta$ are the regularization parameters that control the relative importance of normal and offset consistency, respectively. This loss function ensures that the geometries in consecutive frames are coherently aligned and decomposes the supervision to rotational and translational components of the camera poses. We found that, by adjusting $\lambda_n$ and $\lambda_\delta$, we can achieve more robust pose estimation in indoor environments. To further enhance robustness, we also extract the edge map based on input frame, which is used as a mask to restrict this constraint only to planar regions, while avoiding its effect on the plane edges. Further detail can be found in the supplementary materials.

**Overall global optimization.** By combining the objective functions mentioned above, the overall optimization process is formulated by summing over all the frames:

$$\mathcal{T}, \mathcal{G} = \arg\min_{\mathcal{T}, \mathcal{G}} \sum_{i=1}^{L} \lambda_{\text{plane}}\mathcal{L}_{\text{plane}}^i + \lambda_{\text{rgb}}\mathcal{L}_{\text{rgb}}^i + \lambda_{\text{depth}}\mathcal{L}_{\text{depth}}^i, \tag{11}$$

where the weights $\lambda_{\text{plane}}$, $\lambda_{\text{rgb}}$, and $\lambda_{\text{depth}}$ balance different terms.

### 3.2.2 Alternating Optimization Strategy

Although Eq. 11 defines a unified objective over both the global camera poses $\mathcal{T}$ and the 3D Gaussian parameters $\mathcal{G}$, jointly optimizing them often results in unstable convergence due to gradient interference between the two parameter spaces [21, 6, 14]. To mitigate this issue, we adopt an alternating optimization strategy, updating camera poses and 3D Gaussian parameters in turn. This decoupled scheme improves convergence stability and preserves global consistency, as validated by our ablation studies in Sec. 4.4. Further implementation details, including the gradient update formulation for camera pose parameters, are provided in the supplementary material.

Table 2: Novel view synthesis performance metrics comparison on Replica [44] dataset. For fair comparison, each baseline is trained using its publicly released code and original hyperparameter settings, and evaluated under the same protocol. The best results are highlighted in bold.

| Scenes | Ours | | | CF-3DGS [10] | | | NoPoSplat [61] | | | SelfSplat [22] | | | Nope-NeRF [5] | | | BARF [2] | | | NeRFmm [54] | | |
|---|---|---|---|---|---|---|---|---|---|---|---|---|---|---|---|---|---|---|---|---|---|
| | PSNR↑ | SSIM↑ | LPIPS↓ | PSNR | SSIM | LPIPS | PSNR | SSIM | LPIPS | PSNR | SSIM | LPIPS | PSNR | SSIM | LPIPS | PSNR | SSIM | LPIPS | PSNR | SSIM | LPIPS |
| office0 | **34.32** | **0.92** | **0.09** | 28.50 | 0.87 | 0.28 | 17.34 | 0.57 | 0.47 | 16.50 | 0.56 | 0.35 | 28.26 | 0.85 | 0.51 | 25.23 | 0.78 | 0.48 | 23.52 | 0.58 | 0.49 |
| office1 | **31.78** | **0.92** | **0.10** | 29.11 | 0.85 | 0.18 | 18.52 | 0.41 | 0.41 | 18.11 | 0.59 | 0.40 | 27.35 | 0.69 | 0.40 | 26.38 | 0.63 | 0.53 | 25.33 | 0.61 | 0.53 |
| office2 | **30.77** | **0.91** | **0.11** | 24.97 | 0.78 | 0.29 | 18.54 | 0.56 | 0.34 | 15.97 | 0.45 | 0.43 | 26.77 | 0.76 | 0.35 | 26.52 | 0.73 | 0.43 | 23.28 | 0.61 | 0.54 |
| office3 | **32.37** | **0.90** | **0.12** | 23.70 | 0.75 | 0.25 | 15.84 | 0.54 | 0.43 | 21.70 | 0.57 | 0.45 | 26.01 | 0.74 | 0.41 | 26.37 | 0.61 | 0.41 | 25.25 | 0.78 | 0.45 |
| office4 | **32.12** | **0.90** | **0.09** | 27.49 | 0.78 | 0.28 | 18.93 | 0.59 | 0.35 | 15.49 | 0.56 | 0.38 | 27.64 | 0.84 | 0.26 | 26.48 | 0.72 | 0.36 | 24.08 | 0.70 | 0.43 |
| room0 | **33.21** | **0.89** | **0.06** | 22.60 | 0.68 | 0.38 | 19.96 | 0.47 | 0.45 | 18.63 | 0.54 | 0.47 | 25.33 | 0.72 | 0.38 | 25.53 | 0.64 | 0.32 | 22.93 | 0.48 | 0.51 |
| room1 | **32.13** | **0.87** | **0.08** | 24.50 | 0.74 | 0.35 | 23.92 | 0.53 | 0.46 | 22.15 | 0.48 | 0.43 | 29.42 | 0.80 | 0.38 | 25.54 | 0.56 | 0.54 | 25.40 | 0.69 | 0.43 |
| room2 | **30.40** | **0.88** | **0.10** | 25.34 | 0.74 | 0.34 | 22.08 | 0.55 | 0.36 | 23.42 | 0.55 | 0.48 | 28.96 | 0.61 | 0.47 | 24.83 | 0.53 | 0.60 | 24.16 | 0.43 | 0.40 |
| mean | **32.14** | **0.90** | **0.09** | 25.40 | 0.77 | 0.29 | 19.39 | 0.52 | 0.41 | 18.90 | 0.54 | 0.42 | 27.46 | 0.75 | 0.40 | 25.86 | 0.65 | 0.46 | 24.24 | 0.61 | 0.47 |

# 4 Experiment

## 4.1 Experimental Setup

**Datasets.** We evaluate our method on three public datasets: Replica [44], ScanNet [9], and Tanks & Temples [27], covering both synthetic and real-world scenes. The Replica [44] dataset offers high-fidelity synthetic indoor scenes with precise ground-truth camera poses. Its large textureless regions and complex camera trajectories make it well-suited for evaluating camera pose estimation and novel view synthesis. The ScanNet [9] dataset consists of real-world RGB-D indoor scenes captured in unconstrained environments, presenting challenges such as sensor noise and motion blur. Tanks & Temples [27] dataset contains texture-rich scenes with relatively controlled camera motions. Most previous methods are evaluated on this dataset. We include it to test the generalizability of our method.

**Metrics.** Following prior works [10, 5], we evaluate our method on two key tasks: camera pose estimation and novel view synthesis. For camera pose estimation, we use standard visual odometry metrics [29, 45], including Absolute Trajectory Error (ATE) and rotational Relative Pose Error ($RPE_r$) and translational Relative Pose Error ($RPE_t$). For novel view synthesis, we use standard image quality metrics: Peak Signal-to-Noise Ratio (PSNR), Structural Similarity Index (SSIM) [52], and Learned Perceptual Image Patch Similarity (LPIPS) [66].

**Implementation Details.** Our method is implemented using PyTorch [39], building on the optimization settings from 3DGS [25] with necessary adjustments. A key feature is synchronizing new frame additions with point densification intervals to ensure steady scene expansion. For detailed information, please refer to the supplementary materials.

## 4.2 Replica

In this subsection, we conduct a comparative analysis of our method against several established baselines, including NoPoSplat [61], SelfSplat [22], CF-3DGS [10], Nope-NeRF [5], BARF [2] and NeRFmm [54], all of which are widely recognized for their contributions to camera pose estimation and novel view synthesis.

For camera pose estimation, the optimized camera poses are aligned with Procrustes Analysis, as described in prior works [5, 54], and compared against the ground-truth poses from training views. Quantitative results are summarized in Tab. 1, where our method achieves performance superior to current state-of-the-art approaches.

For novel view synthesis, we adopt the evaluation protocol of CF-3DGS [10] and NeRFmm [54]. The optimized 3DGS model, trained exclusively on the training views, is kept fixed, while the camera poses of the test views are refined by minimizing the photometric reconstruction error between synthesized and ground-truth images. As reported in Tab. 2, our method consistently outperforms all baselines. Moreover, the qualitative comparisons in Fig. 4 highlight that our synthesized images preserve finer details and exhibit sharper textures than those produced by existing methods.

Table 3: Camera pose estimation and novel view synthesis performance metrics comparison on ScanNet [9] dataset. For fair comparison, each baseline is trained using its publicly released code and original hyperparameter settings, and evaluated under the same evaluation protocol. The best results are highlighted in bold.

| Methods | $RPE_t$ ↓ | $RPE_r$ ↓ | ATE ↓ | PSNR ↑ | SSIM ↑ | LPIPS ↓ |
|---|---|---|---|---|---|---|
| NeRFmm [54] | 1.153 | 0.963 | 0.123 | 15.50 | 0.59 | 0.53 |
| BARF [2] | 1.145 | 0.894 | 0.134 | 21.42 | 0.54 | 0.45 |
| Nope-NeRF [5] | 0.763 | 0.688 | 0.040 | 22.11 | 0.64 | 0.39 |
| SelfSplat [22] | 0.984 | 0.932 | 0.231 | 18.28 | 0.52 | 0.45 |
| NoPoSplat [61] | 1.167 | 0.875 | 0.124 | 17.28 | 0.54 | 0.67 |
| CF-3DGS [10] | 0.653 | 0.684 | 0.040 | 23.26 | 0.68 | 0.24 |
| Ours | **0.579** | **0.524** | **0.020** | **25.38** | **0.72** | **0.20** |

## 4.3 ScanNet

To further validate the effectiveness of our method in realistic and unconstrained environments, we evaluate it on ScanNet [9] dataset, which consists of real-world indoor scenes. Compared to Replica [44] dataset, ScanNet [9] dataset introduces additional challenges such as sensor noise, cluttered layouts, and motion blur, making it a more demanding benchmark for pose and reconstruction quality. We follow the evaluation protocol of [10, 5] to assess both camera pose estimation accuracy and novel view synthesis quality. As shown in Tab. 3, our method consistently achieves superior results across all evaluation metrics.

## 4.4 Tanks & Temples

While Replica [44] and ScanNet [9] datasets provide indoor scenes, we further evaluate the generalization ability and robustness of our method on Tanks & Temples [27] dataset, which consists of photogrammetric reconstructions of both indoor and outdoor scenes with texture variations. Following standard evaluation protocols established in [10, 5], we assess both camera pose estimation accuracy and novel view synthesis quality. As reported in Tab. 4, our method performs competitively with state-of-the-art approaches across nearly all metrics. Qualitative results are presented in Fig. 5.

Table 4: Camera pose estimation and novel view synthesis performance metrics comparison on Tanks & Temples [27] dataset. All baseline methods are trained with their official implementations and original configurations, and evaluated following the same protocol to ensure fair and consistent comparison. The best results are highlighted in bold.

| Methods | RPE$_t \downarrow$ | RPE$_r \downarrow$ | ATE $\downarrow$ | PSNR $\uparrow$ | SSIM $\uparrow$ | LPIPS $\downarrow$ |
|---|---|---|---|---|---|---|
| NeRFmm [54] | 1.735 | 0.477 | 0.123 | 22.50 | 0.59 | 0.54 |
| BARF [2] | 1.046 | 0.441 | 0.078 | 23.42 | 0.61 | 0.54 |
| Nope-NeRF [5] | 0.080 | **0.038** | 0.006 | 26.34 | 0.74 | 0.39 |
| SelfSplat [22] | 1.046 | 0.489 | 0.094 | 22.42 | 0.58 | 0.56 |
| NoPoSplat [61] | 1.832 | 0.488 | 0.117 | 20.15 | 0.53 | 0.47 |
| CF-3DGS [10] | 0.041 | 0.069 | 0.004 | 31.28 | 0.93 | 0.09 |
| Ours | **0.034** | 0.043 | **0.003** | **31.68** | **0.94** | **0.07** |

## 4.5 Ablation Study

In this section, we analyze the effectiveness of different key components proposed in our pipeline through systematic ablation studies.

**Effectiveness of local neural geometric representation.** To evaluate the contribution of the local neural geometric representation (LNGR), we conduct an ablation study where LNGR is removed and replaced with a naive initialization strategy: the camera pose of each frame is initialized using that of the previous frame, and depth is directly estimated from a monocular foundation model [24] without LNGR refinement.

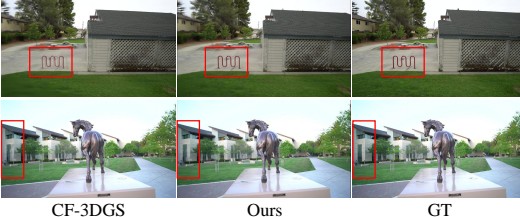

CF-3DGS        Ours        GT

Figure 5: Qualitative results of novel view synthesis on Tanks & Temples [27] dataset. Our NopeRoomGS produces more realistic rendering results than other baselines.

As shown in Tab. 5 (row "w/o LNGR") and Fig. 6, this configuration causes a noticeable drop in both pose accuracy and rendering quality, which underscores the essential role of LNGR $\mathcal{F}_\theta$ in handling challenging indoor scenarios with textureless regions and abrupt camera motion. As described in Sec. 3.1, by jointly optimizing depth and pose across short video clips, $\mathcal{F}_\theta$ produces consistent and robust local scene geometry and camera pose. These outputs provide reliable initializations for local 3D Gaussians, which in turn supply high-quality geometry and pose priors for the subsequent global optimization stage. Such strong initializations are crucial for ensuring stable convergence and achieving high reconstruction fidelity.

**Effectiveness of piecewise planarity assumption.** To assess the impact of the piecewise planarity assumption (PPA), we ablate the PPA constraint from the global optimization stage and replace it with pairwise depth consistency loss computed via reprojection between adjacent frames.

Table 5: Ablation study on Replica [44] dataset. A comparison of our full pipeline and variants without Local Neural Geometric Representation (LNGR), Piecewise Planarity Assumption (PPA), Alternating Optimization Strategy (AOS), and Depth Alignment Loss (DAL) (described in Eq. 7), respectively.

| Methods | RPE$_t \downarrow$ | RPE$_r \downarrow$ | ATE $\downarrow$ | PSNR $\uparrow$ | SSIM $\uparrow$ | LPIPS $\downarrow$ |
|---|---|---|---|---|---|---|
| ours | 0.148 | 0.126 | 0.009 | 32.14 | 0.90 | 0.09 |
| w/o LNGR | 1.532 | 7.422 | 0.063 | 14.82 | 0.65 | 0.42 |
| w/o PPA | 0.192 | 0.179 | 0.023 | 30.34 | 0.89 | 0.14 |
| w/o AOS | 0.205 | 0.189 | 0.019 | 28.28 | 0.89 | 0.15 |
| w/o DAL | 0.156 | 0.158 | 0.078 | 31.42 | 0.90 | 0.14 |

As shown in Tab. 5 (row "w/o PPA") and Fig. 6, this substitution results in a decline in both camera pose accuracy and novel view synthesis quality, thereby substantiating the efficacy of PPA in guiding global optimization. As detailed in Sec. 3.2.1, PPA improves camera pose estimation in textureless regions by introducing geometry-

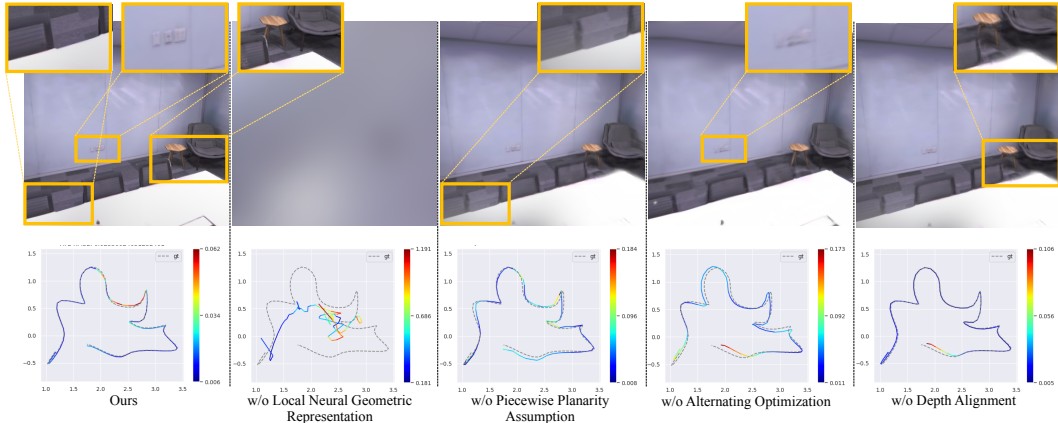

Figure 6: Ablation study for camera pose estimation and novel view synthesis on Replica [44] dataset. We compare our full pipeline with the variants without local neural geometric representations, piecewise planarity assumption, alternating optimization strategy, and depth alignment loss as described in Eq. 7, respectively. The 3D trajectories are projected onto the XY plane of the coordinate system.

aware supervision, and stabilizes optimization under abrupt camera motion by decomposing the supervision signal into rotational (normal alignment) and translational (offset consistency) components, which contributes to the robustness of the overall reconstruction process.

**Effectiveness of alternating optimization strategy.** To evaluate the impact of our incremental and Alternating Optimization Strategy (AOS), we replace it with a joint global optimization method. As shown in Tab. 5 (row "w/o AOS") and Fig. 6, this substitution leads to a drop in both camera pose accuracy and novel view synthesis quality, highlighting the effectiveness of our proposed design. Given the same reliable initialization, alternating optimization plays a critical role in achieving stable convergence and higher reconstruction quality by mitigating interference between camera pose and 3D Gaussian parameters updates.

**Effectiveness of depth alignment loss.** To assess the effectiveness of the Depth Alignment Loss (DAL) in Eq. 7, we replace the optimized depth from the local stage with monocular predictions from a pretrained foundation model [24] to supervise the depth rendered from the global 3DGS stage. As shown in Tab. 5 (row "w/o DAL") and Fig. 6, this leads to degraded camera pose accuracy and novel view synthesis quality. These results highlight the importance of depth alignment supervision and demonstrate the effectiveness of our local neural geometric representation in recovering consistent scene geometry.

## 5   Conclusion and Limitation

In this work, we propose NopeRoomGS, a fully pose-free (i.e., no pose priors) 3D Gaussian Splatting framework that progressively recovers both camera poses and 3DGS representation via a local-to-global optimization paradigm. In the local stage, we introduce a lightweight local neural geometric representation that is jointly optimized on short video clips under supervision from multi-frame tracking and foundation-model depth priors, thereby enabling accurate reconstruction in textureless regions and under abrupt camera motion. In the global stage, we fuse the resulting local 3D Gaussians into a unified 3DGS representation through alternating optimization, ensuring geometric consistency under challenging conditions. Extensive experiments on public datasets demonstrate that our method achieves state-of-the-art performance in both camera pose estimation and novel view synthesis, extending the applicability of 3DGS to real-world, unconstrained environments.

Despite the improved robustness in indoor scenarios and better handling of complex camera motion, several limitations remain. First, the local stage introduces additional computational overhead compared with vanilla 3DGS. Second, even with local-to-global optimization and multiple loss constraints, the method can fail under extremely rapid camera motion or severely sparse inputs. We aim to address these issues in future work.

**Societal Impact.** This technology can benefit AR/VR, robotics, digital content creation, telepresence, and cultural heritage preservation. However, its computational demands may contribute to a higher carbon footprint.

**Acknowledgment.** This study was supported in part by National Key R&D Program of China Project 2022ZD0161100, in part by the Centre for Perceptual and Interactive Intelligence, a CUHK-led InnoCentre under the InnoHK initiative of the Innovation and Technology Commission of the Hong Kong Special Administrative Region Government, in part by NSFC-RGC Project N_CUHK498/24, and in part by Guangdong Basic and Applied Basic Research Foundation (No. 2023B1515130008, XW).

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
