# OpenReview forum: "NopeRoomGS: Indoor 3D Gaussian Splatting Optimization without Camera Pose Input"
_NeurIPS.cc/2025/Conference — NeurIPS 2025 poster_

### Official Review · Reviewer_AuKd · 2025-06-19

**Clarity:** 2
**Significance:** 3
**Originality:** 2
**Rating:** 4
**Confidence:** 4

**Summary:**

This paper presents NopeRoomGS, a novel *pose-free* 3D Gaussian Splatting (3DGS) framework that overcomes the limitations of traditional 3DGS methods, which rely heavily on Structure-from-Motion (SfM) for accurate camera pose estimation. SfM often fails in indoor environments with textureless surfaces or abrupt camera motions. NopeRoomGS addresses these challenges using a local-to-global optimization strategy. In the local stage, it bootstraps reliable 3D Gaussians from short video clips using a lightweight neural geometric representation, guided by multi-frame tracking and depth priors from foundation models. In the global stage, it fuses the local Gaussians and jointly optimizes both camera poses and Gaussian parameters using an alternating optimization scheme to reduce gradient conflicts. Additionally, it enhances robustness by decomposing camera optimization under a piecewise planar assumption. The method achieves state-of-the-art results on multiple benchmarks in both pose estimation and view synthesis.

**Questions:**

1. Would it be possible to include a comparison with methods such as VGG-SfM, VGG-T, or DROID-SLAM, particularly in terms of pose accuracy? Notably, VGG-SfM also employs multi-frame tracking constraints similar to those used in this work.



2. Did the authors observe any failure cases during the experiments?

**Ethical Concerns:**

["NO or VERY MINOR ethics concerns only"]

**Final Justification:**

Thank you to the authors for carefully considering my review comments and providing detailed responses. I believe the authors’ replies have basically addressed my concerns. It was my oversight not to provide references [1] and [2], but since the authors have supplemented comparison experiments with MonoGS and DROID-SLAM and demonstrated the advantages of the current method, I consider this part of my doubts to be resolved. The authors’ explanation regarding the motivation of LNGR is convincing to me, and the experiments also validate the rationality of the LNGR optimization representation. I think this aspect is both innovative and insightful, and the authors should include the corresponding ablation studies and analysis in the main text. In summary, I consider this paper technically solid and possessing a certain degree of novelty, so I recommend (weakly) accepting it.

[1]Matsuki H, Murai R, Kelly P H J, et al. Gaussian splatting slam[C]//Proceedings of the IEEE/CVF Conference on Computer Vision and Pattern Recognition. 2024: 18039-18048. [2] Zhang W, Cheng Q, Skuddis D, et al. HI-SLAM2: Geometry-Aware Gaussian SLAM for Fast Monocular Scene Reconstruction[J]. arXiv preprint arXiv:2411.17982, 2024.

**Limitations:**

yes

**Paper Formatting Concerns:**

The formatting of Table 3 and Table 4 in the paper is reversed.

**Quality:**

2

**Strengths And Weaknesses:**

Streath:
1. The paper is well-structured and clearly written, making it easy for readers to understand and follow.

2. This paper targets the poor performance of existing pose-free 3DGS frameworks in indoor textureless regions and under abrupt camera motion, and proposes a solution that demonstrates superior accuracy and robustness compared to existing methods.



Weakness:
1.  Lack of discussion and comparison with monocular Gaussian SLAM methods. Existing monocular Gaussian SLAM[1][2] approaches are also capable of recovering camera poses and reconstructing 3D Gaussian models using only RGB input. Based on the results reported in this paper, the proposed method does not seem to offer an advantage in terms of rendering quality or pose accuracy compared to monocular Gaussian SLAM[2].
2.  The significance and ablation of the local neural geometric representation are not well justified. The paper employs a lightweight depth estimation network as a parametric representation of geometry, but does not provide a clear explanation of the rationale behind this design choice. Moreover, the ablation study on this aspect is incomplete. The authors should compare the results obtained using optimizable per-pixel depth variables (initialized with the predictions from the depth estimation network). Only through such comparison can the benefit of using a network to parameterize depth be properly demonstrated.
3.  Lack of efficiency reporting. The paper should report optimization time and memory consumption, especially the overhead introduced by using a network to parameterize depth.

---

> ### Author Rebuttal · Authors · 2025-07-31
>
> # Reviewer AuKd
>
> Dear Reviewer AuKd,
>
> We thank you for acknowledging our method’s effectiveness in tackling key challenges in pose-free 3DGS, particularly under textureless scenes and abrupt camera motion. We address your concerns below.
>
> **Q-1**: **The paper lacks discussion and comparison with monocular Gaussian SLAM methods. Existing approaches [1][2] also recover camera poses and 3D Gaussians from RGB input. Based on reported results, the proposed method does not clearly outperform monocular Gaussian SLAM [2] in rendering quality or pose accuracy.**
>
> **A-1:** We sincerely thank the reviewer for raising this important point.
>
> **We notice that the specific literatures you refer to with [1] [2] are not provided in the review.**
>
> We assume they are works similar to the recent monocular Gaussian SLAM methods such as GS3LAM [a] and MonoGS [b]. As far as we know, most of the Gaussian SLAM are RGB-D based, which assume accurate depth inputs for camera pose estimation. However, considering the depth measurements are usually not available in daily settings, **our method only takes RGB as input which is more challenging**.  The pose-free GS and Gaussian SLAM are two related but different tracks of research. In the following, we will elaborate on the difference between them and add more comparisons with the Gaussian SLAM works.
>
> 1. **Distinction from Gaussian SLAM Methods.**
>
>     While both paradigms reconstruct 3D scenes and estimate camera poses from monocular RGB inputs using Gaussian Splatting, they differ fundamentally in their assumptions, application domains, and optimization strategies.
>
>     **(1). Different assumptions and application domains.**
>
>     3DGS-based SLAM methods are designed primarily for online or real-time applications. They assume sequential input with small inter-frame motion and often leverage loop closure mechanisms to ensure long-term consistency. These methods often initialize the pose of the newly arrived frame using the previous frame's pose, heavily relying on temporal continuity.
>
>     In contrast, pose-free 3DGS methods, including ours (NopeRoomGS), are mainly designed for offline novel view synthesis and can operate on sequential video input or unordered image collections.
>
>     Our NopeRoomGS does not rely on temporal ordering for pose initialization. It is specifically designed to handle challenging scenarios with abrupt camera motion or textureless regions, where traditional SLAM-based methods often fail to initialize or maintain accuracy.
>
>     **(2). Different application scenarios: localization vs. photorealistic rendering.**
>
>     The primary objective of SLAM-based 3DGS methods is accurate camera localization and online mapping. As such, their pipelines are optimized to pursue accurate pose/map estimation, with rendering quality being a secondary concern.
>
>     In contrast, the pose-free 3DGS methods like ours focus on producing high-fidelity novel view synthesis alongside robust camera pose estimation.
>
>     In practice, the optimization of camera poses and rendering fidelity could induce gradient conflicts and compromise each other. In our method, we found that the proposed alternating optimization strategy in the global optimization stage can mitigate this issue, resulting in more robust pose estimation and superior rendering quality.
>
> 2. **Additional Comparison with Gaussian SLAMs**
>
>     As you forgot to explicitly mention the method you want us to compare with, we select three recent competitive monocular SLAM baselines: GS3LAM [a], MonoGS[b], and SGS-SLAM [c], and compare our method with them on the Replica dataset using monocular RGB input for fairness. As shown in Table R4-1, our method outperforms other methods in all metrics for rendering quality and pose accuracy, demonstrating the strength of our optimization design.
>
>
> *Table R4-1: Comparison of methods in terms of camera pose accuracy and rendering quality on the Replica dataset*
>
> | Method | ATE ↓ | PSNR ↑ | SSIM ↑ | LPIPS ↓ |
> | --- | --- | --- | --- | --- |
> | GS3LAM | 0.043 | 24.35 | 0.69 | 0.36 |
> | SGS-SLAM | 0.037 | 25.45 | 0.70 | 0.33 |
> | MonoGS | 0.039 | 27.40 | 0.73 | 0.29 |
> | Ours | **0.009** | **32.14** | **0.90** | **0.09** |
>
> We appreciate the reviewer’s suggestion and will include this discussion and comparative results in the revised manuscript to better contextualize our contributions.
>
> Ref:
>
> [a]. GS3LAM: Gaussian Semantic Splatting SLAM.
>
> [b]. MonoGS: Gaussian Splatting SLAM.
>
> [c]. SGS-SLAM: Semantic Gaussian Splatting For Neural Dense SLAM
>
> **Q-2:** **Further explain the motivation for using a neural depth network. Moreover, a comparison with per-pixel optimizable depth (initialized from the same network) is needed to justify this design choice.**
>
> **A-2**:  We sincerely thank the reviewer for the insightful question. Below, we elaborate on the motivation for our Local Neural Geometric Representation (LNGR) and discuss the advantages over a per-pixel optimizable depth design.
>
> 1. **Motivation of LNGR**
>
>     **(1) Ensuring Geometric Consistency Across Frames.**
>
>     LNGR provides a shared depth representation across all frames in a clip, which encourages multi-view geometric consistency and helps reduce scale drift (a common issue in frame-wise optimization pipelines like CF-3DGS). This global structure is difficult to enforce when depths are independently optimized per frame.
>
>     **(2) Improving Optimization Stability and Efficiency.**
>
>     Directly optimizing dense per-frame depth maps introduces millions of parameters (e.g., >6M variables for 20 frames at 640×480 resolution), which tends to pose Jacobian deficiency, especially in textureless regions where valid visual constraints are limited.
>
>     In contrast, our LNGR leverages strong depth priors from a pretrained foundation model and only fine-tunes the final prediction layers (~30K trainable parameters), improving robustness while avoiding overfitting, especially in ill-conditioned regions (e.g., flat walls or low-texture areas) where visual constraints are sparse.
>
> 2. **Ablation: LNGR vs. Per-Pixel Depth Optimization**
>
>     To validate the effectiveness of LNGR, we implemented a baseline variant that directly optimizes per-pixel depth maps initialized from the same pretrained depth network. We refer to this baseline as “Ours (Pixel-wise Depth Optimization)”.
>
>     As shown in Table R4-2, this baseline consistently yields larger pose errors and lower novel view synthesis quality. These results empirically validate the advantages of our LNGR design. We will include the full ablation results in the final version of the paper.
>
>
> *Table R4-2: Comparison of methods in terms of camera pose metrics and rendering quality*
>
> | Method | ATE ↓ | PSNR ↑ | SSIM ↑ | LPIPS ↓ |
> | --- | --- | --- | --- | --- |
> | Ours (Pixel-wise Depth Optimization) | 0.049 | 28.46 | 0.82 | 0.34 |
> | Ours (LNGR) | 0.009 | 32.14 | 0.90 | 0.09 |
>
> **Q-3:** **Lack of efficiency reporting. The paper should report optimization time and memory consumption, especially the overhead introduced by using a network to parameterize depth.**
>
> **A-3**:  Thank the reviewer for this valuable suggestion.
>
> To address this point, we provide a detailed comparison of runtime and memory usage with representative prior methods, including Nope-NeRF  and CF-3DGS .
>
> Our full local-to-global optimization pipeline takes ~81 minutes for each scene running on an RTX 4090 GPU, with peak memory usage <12 GB. The local stage brings ~10 minutes overhead while the global stage takes ~71 minutes.
>
> We provide a direct comparison with Nope-NeRF and CF-3DGS under the same 100-frame sequence setting. As shown in the Table R4-3, our method strikes a favorable balance between runtime and reconstruction quality. While slightly slower than CF-3DGS, our approach offers improved robustness in challenging indoor scenes and is significantly more efficient than Nope-NeRF in terms of both training/rendering efficiency and scalability.
>
> *Table R4-3: Comparison of methods in terms of overhead*
>
> | Method | Time(min) | Memory(GB) |
> | --- | --- | --- |
> | NopeNeRF | ~400 | 6 |
> | CF3DGS | ~63 | 9 |
> | Ours | ~81 | 12 |
>
> **Q-4: Would it be possible to include a comparison with methods such as VGG-SfM, VGG-T, or DROID-SLAM, particularly in terms of pose accuracy?**
>
> **A-4**: We thank the reviewer for the valuable suggestion.
>
> While our method and DROID-SLAM, VGG-SfM, and VGGT all aim to estimate camera poses, they differ fundamentally in formulation and assumptions. However, we fully agree that comparing with these methods, especially in pose accuracy, will strengthen our work.
> We incorporated comparisons with VGG-SfM, VGGT, and DROID-SLAM on Replica dataset. As shown in Table R4-4, our method achieves consistently higher pose accuracy, further validating the advantages of our design.
>
> *Table R4-4:  Comparison of methods in terms of camera pose metrics on Replica dataset*
>
> | Method | RPE_t ↓ | RPE_r ↓ | ATE ↓ |
> | --- | --- | --- | --- |
> | VGG-SfM | 0.547 | 2.003 | 0.024 |
> | VGG-T | 0.521 | 4.547 | 0.013 |
> | DROID-SLAM | 0.783 | 5.234 | 0.054 |
> | Ours | 0.148 | 0.126 | 0.009 |
>
> **Q-5: Did the authors observe any failure cases during the experiments?**
>
> **A-5**: While our method demonstrates strong performance across diverse indoor and outdoor scenes (as shown in Table 1–4 of the paper), our system may fail under extremely challenging conditions. Specifically, as mentioned in the conclusion (Line 376–380 in the manuscript), our method may struggle in scenarios with extremely fast camera motion or overly sparse inputs, where both geometric consistency and photometric supervision become unreliable. We will include additional qualitative examples of such failure cases in the final version to provide a more comprehensive picture. Improving robustness under such challenging conditions is an important direction for future work.

---

> > ### Comment · Reviewer_AuKd · 2025-08-03
> > **My doubts have been resolved**
> >
> > Thank you to the authors for carefully considering my review comments and providing detailed responses. I believe the authors’ replies have basically addressed my concerns. It was my oversight not to provide references [1] and [2], but since the authors have supplemented comparison experiments with MonoGS and DROID-SLAM and demonstrated the advantages of the current method, I consider this part of my doubts to be resolved. The authors’ explanation regarding the motivation of LNGR is convincing to me, and the experiments also validate the rationality of the LNGR optimization representation. I think this aspect is both innovative and insightful, and the authors should include the corresponding ablation studies and analysis in the main text. In summary, I consider this paper technically solid and possessing a certain degree of novelty, so I recommend (weakly) accepting it.
> >
> > [1]Matsuki H, Murai R, Kelly P H J, et al. Gaussian splatting slam[C]//Proceedings of the IEEE/CVF Conference on Computer Vision and Pattern Recognition. 2024: 18039-18048.
> > [2] Zhang W, Cheng Q, Skuddis D, et al. HI-SLAM2: Geometry-Aware Gaussian SLAM for Fast Monocular Scene Reconstruction[J]. arXiv preprint arXiv:2411.17982, 2024.

---

> > > ### Author Response · Authors · 2025-08-03
> > > **Thank you for your positive final rating**
> > >
> > > Dear Reviewer AuKd,
> > >
> > > Thank you for your positive final rating. We’re glad to know that our rebuttal helped clarify our contributions and address your concerns.
> > >
> > > We truly appreciate your thoughtful review and the time you dedicated to our submission.
> > >
> > > Best regards,
> > >
> > > Authors

---

> ### Author Response · Authors · 2025-08-02
>
> Dear Reviewer AuKd,
>
> Thank you again for your thoughtful and constructive review. We would like to kindly draw your attention to our detailed rebuttal, in which we address each of your points with additional analysis and new experimental results, particularly regarding Q-1, Q-2 and Q-3.
>
> In Q-1, you requested comparison with monocular Gaussian SLAM methods ([1], [2]) but did not include the full citations. We’ve done our best to infer the intended works (e.g., GS3LAM, MonoGS, SGS-SLAM), and have provided comprehensive comparisons under fair RGB-only settings (see Table R4-1), where our method outperforms others in both pose accuracy and rendering quality. We hope this addresses your concerns effectively.
>
> Additionally, in Q-2 and Q-3, we further clarified the motivation and unique role of LNGR in our pose-free 3DGS setting, supported by ablations (Table R4-2) and visual analysis (Fig. 6, Table 5).
>
> We sincerely hope these additions help clarify the contributions and improvements brought by our method. If the responses resolve your concerns, we would greatly appreciate your consideration in updating the score.
>
> Best regards,
>
> Submission2437 Authors

---

> > ### Author Response · Authors · 2025-08-03
> >
> > Dear Reviewer AuKd,
> >
> > We hope you're doing well. We’d like to kindly follow up on our earlier author comment and rebuttal. If you have any remaining concerns about our submission, we would be grateful for your input during the discussion phase.
> >
> > We sincerely appreciate your time and thoughtful feedback, and we remain happy to further clarify any points as needed.
> >
> > Best regards,
> >
> > The Authors of Submission2437

---

### Official Review · Reviewer_Ht6p · 2025-06-20

**Clarity:** 3
**Significance:** 2
**Originality:** 2
**Rating:** 4
**Confidence:** 3

**Summary:**

Traditional 3D reconstruction methods for indoor scenes often rely on Structure from Motion to estimate camera trajectories and scene geometry. NopeRoomGS introduces a new approach that avoids this dependency entirely. It uses a two-stage local-to-global optimization framework. In the first stage, it combines a pretrained monocular depth model like RepDepth with a visual tracker such as CoTracker to estimate local geometry and camera poses. In the second stage, it refines the entire Gaussian scene representation by enforcing multiview consistency and applying structural constraints such as planar surfaces. Experiments on standard indoor datasets like ScanNet and Replica show that NopeRoomGS outperforms existing methods including CF-3DGS and NopeNeRF in reconstruction quality.

**Questions:**

1. The method relies on pretrained models like RepDepth and CoTracker, which carry strong geometric and motion priors. It’s unclear whether this still qualifies as “pose-free” under your definition. Please clarify.

2. The appendix includes ablation results without RepDepth and CoTracker, but it would be helpful to test their replaceability and robustness. Try using simpler or non-learned alternatives, or simulate degraded performance, to better understand how sensitive the system is to prior quality.

3. Please consider releasing the code, or at least providing an anonymous test script. Also, include a table comparing resource usage (e.g., memory, training time) with CF-3DGS on the same datasets, and indicate where the main computational bottlenecks are.

4. The local-to-global structure is said to improve robustness in low-texture and fast-motion scenarios, but this isn’t well explained. Some theoretical reasoning or targeted experiments would help clarify why this setup works better under such conditions.

**Ethical Concerns:**

["NO or VERY MINOR ethics concerns only"]

**Final Justification:**

My questions have been satisfactorily addressed. The paper is technically solid and contains a certain degree of innovation.

**Limitations:**

yes

**Paper Formatting Concerns:**

Tables 2, 3, 5 and Figure 5 are left-aligned with text on the right, creating a split-column look. It breaks the flow—suggest adjusting layout for better readability.

**Quality:**

3

**Strengths And Weaknesses:**

This work proposes a method for 3D Gaussian Splatting without relying on SfM or ground-truth poses. The pipeline is clear and well-structured, combining monocular depth and tracking for local geometry, followed by global pose refinement using structural priors. The design is practical, and the method outperforms several pose-free baselines. Some ablation results support the effectiveness of key components.

However, the term “pose-free” is questionable. The method depends on pretrained models like RepDepth and CoTracker, which encode strong geometric priors. It would be more accurate to describe it as weakly supervised rather than truly pose-free. The system is also fairly complex, involving multiple non-end-to-end modules. The paper lacks details on resource usage, efficiency, and code availability, which raises concerns about reproducibility and deployment.

---

> ### Author Rebuttal · Authors · 2025-07-31
>
> # Reviewer Ht6p
>
> Dear Reviewer Ht6p,
>
> We sincerely thank you for the detailed and thoughtful review. We appreciate the positive comments on the clarity, structure, and technical soundness of our pipeline. We address the raised concerns below.
>
> **Q-1: The method relies on pretrained models like RepDepth and CoTracker, which carry strong geometric and motion priors. It’s unclear whether this still qualifies as “pose-free” under your definition. Please clarify.**
>
> **A-1:** We sincerely thank the reviewer for highlighting this point.
>
> Consistent with previous work (e.g. CF-3DGS, NopeNeRF), we define pose-free as requiring no pre‑computed camera extrinsics from SfM, SLAM, or IMU. Our pipeline therefore begins with unknown poses and optimises them.
>
> While we do leverage pretrained models, CoTracker and RepDepth (also known as **Marigold**, as it is more commonly cited),  as weak priors during local stage, their outputs are treated as soft guidance, not ground truth, because the depth scale can differ from frame to frame. To obtain a consistent local scene structure, a lightweight Local Neural Geometric Representation (LNGR) is trained to align with, not copy, these noisy cues in the local stage; both depth and tracks are subsequently refined during the global optimization. As neither module supplies external 6‑DoF poses, the entire pipeline remains genuinely SfM‑free.
>
> We hope this clarifies our use of pretrained components and why the method remains within the accepted scope of pose‑free reconstruction.
>
> **Q-2: The appendix includes ablation results without RepDepth and CoTracker, but it would be helpful to test their replaceability and robustness. Try using simpler or non-learned alternatives, or simulate degraded performance, to better understand how sensitive the system is to prior quality.**
>
> **A-2:** We thank the reviewer for this constructive suggestion.
> To evaluate the robustness of our method and the replaceability of different depth and correspondence predictors, we replace the pretrained models: “RepDepth” and CoTracker, with simpler or classical alternatives:
>
> (1) The depth prediction model “RepDepth” ( ～1B parameters) is replaced with MiDaS ( ～20M parameters), a lightweight monocular estimation depth model;
>
> (2) The motion prior CoTracker( ～44 M parameters) is replaced with the classical two-frame optical flow model RAFT ( ～5M parameters).
>
> We evaluated these variants on the Replica dataset. As shown in Table R3-1,  although the performance could slightly degrade with these weaker models, the system still converges robustly and outperforms existing SfM-free baselines(CF-3DGS: 0.059(ATE), 25.40(PSNR), Table 1 and Table 4 of the manuscript). This suggests that our method can robustly work with different types of pretrained foundation models, demonstrating the effectiveness of our proposed modules.
>
> *Table R3-1: Comparison of methods with different model priors on the Replica dataset*
>
> | Method | ATE ↓ | PSNR ↑ | SSIM ↑ | LPIPS ↓ |
> | --- | --- | --- | --- | --- |
> | Ours (CoTracker + Marigold) | 0.009 | 32.14 | 0.90 | 0.09 |
> | Ours (CoTracker + Midas) | 0.018 | 30.23 | 0.90 | 0.09 |
> | Ours (RAFT + Marigold) | 0.020 | 29.14 | 0.89 | 0.10 |
> | Ours (RAFT + Midas) | 0.024 | 29.79 | 0.83 | 0.15 |
>
> **Q-3**: **Please consider releasing the code, or at least providing an anonymous test script. Also, include a table comparing resource usage (e.g., memory, training time) with CF-3DGS on the same datasets, and indicate where the main computational bottlenecks are.**
>
> **A-3:** We thank the reviewer for the valuable suggestions regarding reproducibility and transparency.
>
> 1. **Code and Reproducibility.**
>
>     Due to rebuttal guidelines, we are unable to include any external links or non-anonymized scripts at this stage. As noted in the paper, we will publicly release the code and trained models upon acceptance to ensure full reproducibility.
>
> 2. **Resource Usage.**
>
>     In response to the reviewer’s suggestion, we include a table summarizing the resource usage (e.g., memory and training time) and elaborate on the main computational bottlenecks in our pipeline below.
>
>     As shown in Table R3-2, the total runtime of our pipeline is less than 90 minutes per scene on an RTX 4090 GPU, with peak memory usage below 12 GB. The additional time and memory overhead primarily stems from the Local Neural Geometric Representation (LNGR) and the inference of foundation models (RepDepth and CoTracker). While these components introduce modest computational overhead, they significantly improve reconstruction quality, as shown in Table 5 of the paper, where PSNR increases from 14.82 dB to 32.14 dB.
>
>
> *Table R3-2: Comparison of methods in terms of overhead*
>
> | Method | Time (min) | Memory(GB) |
> | --- | --- | --- |
> | NopeNeRF | ~400 | 6 |
> | CF3DGS | ~63 | 9 |
> | Ours | ~81 | 12 |
>
> **Q-4: The local-to-global structure is said to improve robustness in low-texture and fast-motion scenarios, but this isn’t well explained. Some theoretical reasoning or targeted experiments would help clarify why this setup works better under such conditions.**
>
> **A-4:** Thanks for your comment.
>
> 1. **Theoretical Explanation**
>
>     We appreciate the reviewer’s request for clarification regarding the robustness of our local-to-global strategy in low-texture and fast-motion scenarios.
>
>     One of the key distinctions of our framework compared to prior pipelines (e.g., CF-3DGS) lies in how initial camera poses are optimized. Specifically, we segment the video into short, overlapping clips and perform joint pose and geometry optimization at the clip level, rather than frame-by-frame. This clip-wise formulation improves robustness to fast camera motion in two ways:
>
>     **(1). Better-conditioned optimization.**
>
>     Each clip forms a small-scale multi-view reconstruction problem with overlapping views, which improves numerical stability and optimization convergence.
>
>     **(2).Stronger geometric constraints.**
>
>     Multi-view consistency across multiple frames enables more accurate pose disambiguation than pairwise (frame-to-frame) approaches, particularly when motion blur or abrupt motion causes image misalignment.
>
>     In low-texture scenarios, traditional geometric methods often struggle due to a lack of reliable visual features. Our method addresses this by integrating dense depth and correspondence priors from pretrained foundation models (RepDepth and CoTracker) during the local stage. These priors provide global structure-aware cues that are especially beneficial in regions where SIFT/SfM-based methods may fail. The priors act as soft constraints that guide the local optimization even in the absence of strong appearance gradients.
>
> 2. **Quantitative Comparison**
>
>     To better validate the robustness of L2G structure,  we conducted an ablation study, increasing the difficulty of the reconstruction task by downsampling the input video frames at varying intervals (interval steps = 1, 5, 10).
>
>     We compare our full method with a baseline that disables the L2G mechanism (“Ours w/o L2G”), which performs camera pose optimization similar to prior work. As shown in Table R3-3, our method with L2G structure consistently yields lower Absolute Trajectory Error (ATE) and higher rendering quality metrics. These results confirm that the L2G structure significantly improves robustness and accuracy under precisely the challenging conditions described.
>
>
> *Table R3-3: Comparison of methods in terms of robust on Replica in different sample steps.*
>
> | Method | Step | ATE ↓ | PSNR ↑ | SSIM ↑ | LPIPS ↓ |
> | --- | --- | --- | --- | --- | --- |
> | Ours w/o L2G | 1 | 0.032 | 30.32 | 0.77 | 0.21 |
> | Ours w/o L2G | 5 | 0.052 | 24.40 | 0.73 | 0.30 |
> | Ours w/o L2G | 10 | 0.059 | 25.71 | 0.67 | 0.33 |
> | Ours w L2G | 1 | **0.007** | **34.16** | **0.91** | **0.09** |
> | Ours w L2G | 5 | **0.010** | **33.23** | **0.90** | **0.11** |
> | Ours w L2G | 10 | **0.009** | **32.14** | **0.90** | **0.09** |
>
> We will incorporate this explanation and analysis into the revised manuscript to improve clarity and completeness.

---

> ### Author Response · Authors · 2025-08-04
>
> Dear Reviewer Ht6p,
>
> We would like to kindly follow up on our earlier author comment and rebuttal. If you have any remaining concerns or suggestions, we would be truly grateful for your feedback during the discussion phase.
>
> Thank you again for your thoughtful and detailed review. We sincerely appreciate your positive comments on the clarity, structure, and technical soundness of our work. In our rebuttal, we have addressed your key concerns as follows:
>
> - **Pose-free Definition:** We clarified that our method, consistent with prior work (e.g., CF-3DGS, NopeNeRF), remains genuinely pose-free, as it does not require external 6-DoF poses from SfM, SLAM, or IMU. Pretrained depth and tracking models are only used as soft priors and are jointly refined during optimization.
> - **Robustness to Priors:** We tested the system with lighter-weight or classical alternatives (e.g., MiDaS, RAFT), showing that it remains robust and continues to outperform existing pose-free baselines, even with degraded priors.
> - **Reproducibility and Efficiency:** While we cannot share code during rebuttal, we commit to open-sourcing the project upon acceptance. We also included a comparison table of memory and runtime against CF-3DGS and NopeNeRF, and identified LNGR and prior inference as the main contributors to computational cost.
> - **L2G Structure Robustness:** We provided theoretical explanation and new ablations to show that our local-to-global (L2G) formulation enhances robustness under fast motion and low-texture scenarios—challenges where traditional methods often fail.
>
> We remain happy to further clarify any points as needed. Thank you once again for your time and contributions to the review process.
>
> Best regards,
>
> The Authors of Submission2437

---

> > ### Comment · Reviewer_Ht6p · 2025-08-05
> >
> > Thank you for your response, which has addressed most of my concerns. However, I suggest conducting a full ablation study—comparing local only, global only, and local-to-global (L2G) settings—specifically on datasets that include low-texture and fast-motion scenes. These challenging scenarios are essential for rigorously evaluating the robustness and generalization of your method, as benchmarks like Replica may not sufficiently cover them. Such experiments, together with a more detailed theoretical analysis or relevant citations, would significantly strengthen the credibility and impact of your work.

---

> > > ### Author Response · Authors · 2025-08-06
> > >
> > > Dear Reviewer Ht6p,
> > >
> > > We are really glad to hear that we have addressed most of your concerns. Thank you for the follow-up suggestion.
> > >
> > > **Q: I suggest conducting a full ablation study—comparing local only, global only, and local-to-global (L2G) settings—specifically on datasets that include low-texture and fast-motion scenes. These challenging scenarios are essential for rigorously evaluating the robustness and generalization of your method, as benchmarks like Replica may not sufficiently cover them. Such experiments, together with a more detailed theoretical analysis or relevant citations, would significantly strengthen the credibility and impact of your work.**
> > >
> > > **A:**  We appreciate your suggestion for further ablation and analysis.
> > >
> > > 1. We agree that a more comprehensive ablation, including *local only*, *global only*, and *local-to-global (L2G)* configurations, can better highlight the effectiveness of our design.
> > >
> > > 	In Table 5 of the manuscript, we compare the **global-only variant ("w/o LNGR")** with our full pipeline incorporating the local-to-global mechanism ("Ours"). The global-only variant relies on frame-by-frame initialization, similar to CF-3DGS.
> > >
> > >     For the comparison with the **local-only variant**, it is not viable given our current framework, because our local stage only produces per-frame depth maps and initial camera poses. The Gaussian blobs (used for rendering) are initialized and optimized in the global stage. Ablating the global stage requires significant modification of our current framework and making the evaluation infeasible.
> > >
> > > 2. We agree that evaluating on datasets with low-texture and fast-motion scenes can further demonstrate its robustness and generalization. In our manuscript, we have already tested our method on three such datasets: Replica, ScanNet, and Tanks and Temples. But we respectfully disagree with the notion that Replica and ScanNet are not representative enough to cover the challenging low-texture and fast-motion scenes. The choice of Replica and ScanNet is based on the following considerations:
> > >
> > >     **Wide adoption in prior work**: These datasets are widely adopted benchmarks in the literature on indoor scene reconstruction with textureless conditions. Methods such as NICER-SLAM, MonoSDF, MonoInstance, and GSRec (*Surface Reconstruction from 3D Gaussian Splatting via Local Structural Hints*) have all utilized them, making these datasets well-suited for fair and direct comparison.
> > >
> > >     **Challenging scene properties**: The Replica and ScanNet datasets exhibit large textureless regions and frequent abrupt camera motions, making them good dataset candidates to evaluate our method. In Fig. 2 of the manuscript, we quantitatively compare (A) the proportion of low-texture areas, (B) camera trajectory curvature, and (C) inter-frame rotation across Replica, ScanNet, and Tanks and Temples. The results show that Replica and ScanNet contain higher proportions of low-texture areas and more frequent camera motion abruption, which is much more challenging than the Tanks and Temples dataset that is frequently used in prior pose-free approaches. Visual evidence in Fig. 1, Fig. 6 of the manuscript, and Fig. 1 in CF-3DGS also highlights the low-texture nature and abrupt camera motion of these scenes. To evaluate robustness under even more severe camera motion conditions, we also downsample video frames using varying strides (i.e., 1, 5, and 10), as reported in Table R3-3.
> > >
> > >
> > > That said, we welcome suggestions for additional challenging datasets and will consider including them in the final revision.
> > >
> > > If there are any further concerns you'd like us to address, please feel free to let us know. We would be more than happy to respond.
> > >
> > > Best regards,
> > >
> > > The Authors of Submission 2437

---

> ### Author Response · Authors · 2025-08-07
>
> Dear Reviewer Ht6p,
>
> We hope this message finds you well.
>
> We would like to follow up on our previous response regarding your suggestion. We sincerely appreciate your constructive feedback, which has greatly helped us improve our work.
>
> If there are any remaining concerns or additional suggestions you would like us to address, we would be very happy to hear them. Please don’t hesitate to let us know if anything remains unclear or incomplete. We truly value your insights and are committed to further clarifying or extending our work if needed.
>
> Thank you again for your time and thoughtful review.
>
> Best regards,
>
> The Authors of Submission 2437

---

> ### Comment · Reviewer_Ht6p · 2025-08-08
>
> Thank you for the reply, but I still have some doubts. From a stricter definition of “pose-free” — i.e., not using any models trained with posed data that may implicitly encode geometric or motion information — the proposed method may not fully meet this criterion. While RepDepth and CoTracker do not directly output 6DoF camera poses, both are pre-trained on large-scale datasets with camera pose annotations, and their predictions may implicitly provide geometric and motion constraints.
>
> The ablation study in Supplementary Section D.3 (removing LNGR and replacing it with naïve pose propagation and unrefined monocular depth) effectively removes the weak priors from RepDepth/CoTracker. The results show that, without these priors, the method can still run but suffers a notable performance drop. This suggests that weak priors play an important role in convergence stability and final performance.
>
> I remain somewhat uncertain about how this reliance fits within the “pose-free” category under the stricter definition. I would appreciate it if you could discuss these results more explicitly and clarify in your rebuttal how you position and justify your method’s classification as “pose-free” given this dependency.

---

> ### Author Response · Authors · 2025-08-08
>
> Dear Reviewer Ht6p:
>
> Thank you for the question.  We appreciate the opportunity to clarify how *NopeRoomGS* satisfies the “pose-free” criterion.
>
> In the novel view synthesis (NVS) community, the term *“pose-free”*(a.k.a. “pose-unknown” or “COLMAP-free”) conventionally **refers to methods that do not require camera pose inputs** (either estimated poses from SfM/SLAM or ground truth from simulators) when optimizing the scene representation. This definition has been consistently adopted by prominent pose-free NVS works such as NeRF--[1], BARF[2], Nope-NeRF[3], CF-3DGS[4], and more recent approaches like NoPoSplat[5], SelfSplat[6].
>
> **It is also quite common to incorporate geometric priors during the optimization among pose-free methods**. For instance, BARF[2] incorporates a pose prior during optimization, and Nope-NeRF[3] leverages monocular depth priors from a pre-trained model. Similarly, feedforward *pose-free* methods (e.g., NoPoSplat [5], SelfSplat [6], Splatt3r [7], InstantSplat [8]) **directly use geometric priors and are trained on posed datasets**.
>
> Moreover, we also want to clarify that although the datasets on which RepDepth and CoTracker are trained may include pose annotations, their training procedures do not utilize these annotations.
>
> **To further address your concerns, we would like to clarify the Supplementary Section D.3.** In Supplementary Section D.3, the statement *“removing LNGR and replacing it with naïve pose propagation and unrefined monocular depth[7]”* refers to removing LNGR in the optimization of the local stage, but we still rely on the depth predictions from RepDepth to initialize the Gaussian blobs. This ablation study is used to demonstrate the effectiveness of our proposed optimization scheme in the local stage, but it does **not** imply that our performance gains are primarily due to pre-trained foundation models, as the depth prior remains present even in this ablation.
>
> This ablation study, in Supplementary Section D.3, also shows that **consistent camera poses and scene geometry cannot be recovered using only the geometric priors estimated by RepDepth/CoTracker, respectively.**  This is primarily due to the **scale inconsistency** of monocular depth estimation across frames and the lack of global coherence in the correspondence tracking, without **joint optimization of depth and camera pose provided by LNGR**.
>
> To avoid potential misinterpretation and to better reflect that our method does not require camera poses from SfM/SLAM systems or ground-truth simulators, we are also open to using alternative terminology such as “pose-unknown” or “COLMAP-free” in the final version.
>
> Ref:
>
> [1]. NeRF--: Neural Radiance Fields Without Known Camera Parameters
>
> [2]. BARF : Bundle-Adjusting Neural Radiance Fields
>
> [3]. NoPe-NeRF: Optimising Neural Radiance Field with No Pose Prior
>
> [4]. COLMAP-Free 3D Gaussian Splatting
>
> [5]. No Pose, No Problem: Surprisingly Simple 3D Gaussian Splatsfrom Sparse Unposed Images
>
> [6]. SelfSplat: Pose-Free and 3D Prior-Free Generalizable 3D Gaussian Splatting
>
> [7]. Splatt3r:  Zero-shot Gaussian Splatting from Uncalibrated Image Pairs
>
> [8]. Instantsplat: Sparse-view Gaussian Splatting in Seconds
>
>
> Thank you again for your time and review.
>
> Best regards,
>
> The Authors of Submission 2437

---

> > ### Comment · Reviewer_Ht6p · 2025-08-09
> >
> > Thank you for the clarification, my question is answered and I will update my score.

---

> > > ### Author Response · Authors · 2025-08-09
> > > **Thank you for your review and updated score**
> > >
> > > Dear Reviewer Ht6p,
> > >
> > > We sincerely appreciate your follow-up and are glad that our clarification addressed your question.
> > >
> > > Thank you for your time, thoughtful review, and for updating your score.
> > >
> > > Best regards,
> > >
> > > The Authors of Submission 2437

---

### Official Review · Reviewer_Xtpw · 2025-06-20

**Clarity:** 3
**Significance:** 3
**Originality:** 3
**Rating:** 4
**Confidence:** 5

**Summary:**

In this paper, to address the challenge of processing textureless regions and abrupt camera motion in pose-free 3DGS methods, the author proposes NopeRoomGS, a pose-free 3DGS framework with a local-to-global strategy. In detail, the authors split an image sequence into several video clips. For each clip viewed as a local stage, the camera pose and depth are jointly optimized. With the optimized depth, the 3D Gaussians can be initialized. In the global stage, the local 3D Gaussians will be fused into the global set of 3D Gaussians, and then the 3D Gaussians and camera poses will be optimized together in the global stage. The authors introduce a novel pose constraint based on piecewise planarity assumption to achieve more robust pose estimation. The experiments in several datasets show the effectiveness of the proposed framework.

**Questions:**

Please refer to the weakness part.

**Ethical Concerns:**

["NO or VERY MINOR ethics concerns only"]

**Final Justification:**

I have read them carefully, and they address all of my initial comments. Please incorporate these discussions into the final version for completeness. Therefore, I would like to maintain my original positive rating.

**Limitations:**

Yes.

**Paper Formatting Concerns:**

No.

**Quality:**

3

**Strengths And Weaknesses:**

Strengths:
1. The paper is well-written and easy to follow.
2. The authors analyze the distributions of low-texture regions, camera trajectory curvature, and camera rotation angle difference across three benchmark datasets, and then figure out the fundamental challenges of textureless regions and abrupt camera motion. This analysis is meaningful and provides ideas on the bottleneck in the recent pose-free 3DGS methods.
3. A novel pose constraint based on piecewise planarity assumption is proposed to enhance the robustness of camera pose estimation.
4. The local-to-global strategy ensures the local geometric consistency, which leads to a better initialization for 3D Gaussians.

Weakness:
1. For each video clip, the authors use a monocular depth estimator to estimate depth for each image in the local stage. I would like to know how to ensure the optimized depths in the local stage have the same scale as the depths in the global stage, since in the fusing part, the scale of the location of local 3D Gaussians should align with the global 3D Gaussians. We know that the jointly optimized camera poses and depths in the local stage are missing a scale to the real scene, then for different video clips, there will be different scales for each clip. Therefore, I would like to know how to figure out this issue.
2. It would be better to have memory and time comparisons with previous methods.
3. It would be better if the authors could explain the difference between the 3DGS-based SLAM methods and the pose-free 3DGS methods, and provide some quantitative comparisons.

---

> ### Author Rebuttal · Authors · 2025-07-31
>
> # Reviewer Xtpw
>
> Dear Reviewer Xtpw,
>
> Thank you for your thoughtful and professional feedback and highly valuable comments. We sincerely appreciate your recognition of the motivation and methodology underlying our work.
>
> **Q-1:** **For each video clip, the authors use a monocular depth estimator to estimate depth for each image in the local stage. I would like to know how to ensure the optimized depths in the local stage have the same scale as the depths in the global stage.**
>
> **A-1:** We appreciate the reviewer’s thoughtful observation regarding the potential scale-drift issue.
>
> In our framework, each local clip shares at least one frame with its preceding clip, as illustrated in Fig. 2 in our paper. During global fusion, clips are merged sequentially. Suppose we have already fused the first $n$ frames, that is ${1, 2, ..., n}$ ; when merging the next local clip consisting of frames ${n, n+1, ..., n+l}$ (where $l$ is the clip length), frame $n$ serves as the overlapping frame.
>
> In the local optimization stage, we fix the camera pose of the shared frame $n$ to its globally optimized pose from the previous stage. We then jointly optimize the poses and depths of the remaining frames ${n+1, ..., n+l}$, using scale-and-shift-invariant regularization loss, as described in equation(4) of Sec. 3.1. This design ensures that the new clip is aligned in scale with the previously fused global model. As a result, the metric scale is propagated from one clip to the next, maintaining a consistent global scale across the entire sequence.
>
> We will incorporate this explanation into Sec. 3.2 and explicitly reference Fig. 2 to improve clarity.
>
> **Q-2:** **It would be better to have memory and time comparisons with previous methods**
>
> **A-2**: We thank the reviewer for the valuable suggestion. To address this point, we report detailed runtime and memory usage and compare with representative methods, including Nope-NeRF [1] and CF-3DGS [2].
>
> Our full pipeline takes ~81 minutes per scene on an RTX 4090 GPU, with peak memory usage under 12 GB. The local stage adds ~10 minutes overhead (due to LNGR), while the global stage takes ~71 minutes.
>
> As shown in Table R2-1, under the same 100-frame setting, our method offers a favorable trade-off between runtime and reconstruction quality. It is slightly slower than CF-3DGS but achieves greater robustness in challenging indoor scenes, and is significantly more efficient and scalable than Nope-NeRF in both training and rendering.
>
> Table R2-1: Runtime and memory usage comparison on 100-frame sequences
>
> | Method | Time (min) | Memory(GB) |
> | --- | --- | --- |
> | NopeNeRF | ~400  | 6 |
> | CF-3DGS | ~63 | 9 |
> | Ours | ~81 | 12 |
>
> We will incorporate this analysis into the revised manuscript to improve clarity and better contextualize the efficiency of our method.
>
> Reference:
>
> [1]. NoPe-NeRF: Optimising Neural Radiance Field with No Pose Prior
>
> [2]. CF-3DGS: COLMAP-Free 3D Gaussian Splatting
>
> **Q-3: It would be better if the authors could explain the difference between the 3DGS-based SLAM methods and the pose-free 3DGS methods, and provide some quantitative comparisons.**
>
> **A-3:** We sincerely thank the reviewer for this insightful question and the opportunity to clarify the conceptual and practical distinctions between 3DGS-based SLAM methods and pose-free 3DGS reconstruction methods.
>
> While both paradigms reconstruct 3D scenes and estimate camera poses from monocular RGB inputs using Gaussian Splatting, they differ fundamentally in their assumptions, goals, optimization strategies, and robustness.
>
> 1. **Different Assumptions and Application Domains.**
>
>     3DGS-based SLAM methods are designed primarily for online or real-time applications. They assume sequential input with small inter-frame motion and often leverage loop closure mechanisms to ensure long-term consistency. These methods often initialize the pose of the newly arrived frame using the previous frame's pose, heavily relying on temporal continuity.
>
>     In contrast, pose-free 3DGS methods, including ours (NopeRoomGS), are mainly designed for offline novel view synthesis and can operate on sequential video input or unordered image collections.
>
>     Our NopeRoomGS does not rely on temporal ordering for  pose initialization. It is specifically designed to handle challenging scenarios with abrupt camera motion and large textureless regions, where traditional SLAM-based methods often fail to initialize or maintain accuracy.
>
> 2. **Different Optimization Focus: Localization vs. Photorealism**
>
>     The primary objective of SLAM-based 3DGS methods is **accurate camera localization** and **online mapping**. As such, their pipelines are optimized for **pose/map accuracy**, with rendering quality being a secondary concern.
>
>     In contrast, pose-free 3DGS methods like ours focus on producing high-fidelity novel view synthesis alongside robust camera pose estimation. In our method, this is achieved through a carefully designed alternating optimization strategy in the global stage, which decouples the optimization of camera poses and 3D Gaussian parameters to mitigate gradient conflicts. This results in superior rendering fidelity and competitive, often superior, pose accuracy.
>
> 3. **Quantitative Comparison**
>
>     To further clarify the differences, we compare our method with three representative GS-SLAM methods: GS3LAM [1], SGS-SLAM [2], and MonoGS [3], on the Replica dataset, following the same input protocol using only monocular RGB images.
>
>     As shown in Table R2-2, our method significantly outperforms SLAM-based counterparts across all metrics, including absolute trajectory error (ATE) and photometric quality, highlighting the effectiveness of our local-to-global pipeline in challenging indoor environments.
>
>
> *Table R2-2: Comparison of methods in terms of camera pose accuracy and rendering quality on the Replica dataset*
>
> | Method | ATE ↓ | PSNR ↑ | SSIM ↑ | LPIPS ↓ |
> | --- | --- | --- | --- | --- |
> | GS3LAM | 0.043 | 24.35 | 0.69 | 0.36 |
> | SGS-SLAM | 0.037 | 25.45 | 0.70 | 0.33 |
> | MonoGS | 0.039 | 27.40 | 0.73 | 0.29 |
> | **Ours** | **0.009** | **32.14** | **0.90** | **0.09** |
>
> We will incorporate this detailed comparison and explanation into the revised manuscript to provide a clearer understanding of the distinctions and advantages of our method.
>
> Reference:
>
> [1] GS3LAM: Gaussian Semantic Splatting SLAM, ACM MM.
>
> [2] SGS-SLAM: Semantic Gaussian Splatting for Neural Dense SLAM.
>
> [3] MonoGS: Gaussian Splatting SLAM with Monocular RGB Input.

---

> ### Author Response · Authors · 2025-08-01
>
> Dear Reviewer Xtpw,
>
> Thank you again for your constructive and insightful review of our paper. We greatly appreciate your recognition of our method’s motivation, the proposed piecewise planar constraint, and the local-to-global strategy. We're also grateful for your suggestions, which have led us to significantly improve the clarity and completeness of our work.
>
> In our rebuttal, we have:
>
> - **Clarified the scale consistency strategy** during clip fusion using overlapping frames and fixed-pose alignment, as outlined in Fig. 2.
>
> - **Added runtime and memory usage comparisons** with strong baselines (Nope-NeRF and CF-3DGS), highlighting a favorable trade-off.
>
> - **Provided a detailed distinction and quantitative comparison** between SLAM-based 3DGS methods and pose-free 3DGS frameworks, which demonstrates the effectiveness of our approach across pose accuracy and rendering quality metrics.
>
> We hope that these updates address your concerns and further highlight the novelty, clarity, and robustness of our proposed method. If you find the responses satisfactory and the paper's contributions meaningful, we would sincerely appreciate your consideration of a higher score. We look forward to your further input during the discussion phase.
>
> Thank you again for your thoughtful review and your time during the discussion phase.
>
> Best regards,
>
> Submission2437 Authors

---

> > ### Comment · Reviewer_Xtpw · 2025-08-04
> > **I will keep my original positive rating**
> >
> > Thank you for the authors’ detailed responses. I have read them carefully, and they address all of my initial comments. Please incorporate these discussions into the final version for completeness. Therefore, I would like to maintain my original positive rating.

---

### Official Review · Reviewer_Hqtq · 2025-06-23

**Clarity:** 3
**Significance:** 2
**Originality:** 2
**Rating:** 4
**Confidence:** 5

**Summary:**

This paper introduces NopeRoomGS, a pose-free indoor 3D Gaussian Splatting framework that that does not require Structure-from-Motion (SfM)-based camera pose initialization (similar in the settings to Nope-NeRF and Colmap-free 3DGS (CF-3DGS), etc.). The paper proposes a local-to-global optimization strategy to address textureless regions and abrust camera motion. It first estimates local pose and depth from short video clips using a pre-trained depth estimation foundation model, then globally refine these using photometric, depth and geometric objective functions, including a planarity-based pose regularizer. As a result, the paper conducts experiments on 3 public datasets, Tanks and Temples, Replica, and ScanNet to demontrate that the proposed method achieves SOTA performance.

**Questions:**

1. I think it would be helpful to provide evidence to support the pose-free claim.

2. Also, direct comparison vs. GS-based SLAM method and discussion should be added.

3. More insights on alternating strategy would be appreciated.

4. The rest of the major and minor weaknesses.

I do think there are interesting merits in the paper, for example, there are strong performance improvement in Replica.

I would encourage the authors to address the concerns in the rebuttal. I also remain open to the possibility that I may have overlooked certain aspects

**Ethical Concerns:**

["NO or VERY MINOR ethics concerns only"]

**Final Justification:**

Thank you to the authors for the detailed rebuttal. I believe the majority of questions have been clearly addressed.

At this point, I have no major remaining concerns with the paper. However, I would like to offer two comments to conclude my response:

1. The authors clarified that only eight scenes from the Tanks and Temples dataset were evaluated, which is the same protocol being used in some of the previous baselines. This detail (which specific scenes) should be explicitly stated in the final paper to avoid potential confusion. Since the authors have promised to include full dataset results in the camera-ready version, I am leaning towards raising my rating.

2. I also agree with Reviewer Ht6p that the term "pose-free" may feel somewhat strong or idealized, given the current state of the field. While not technically incorrect, it could give readers the impression that the problem is well solved, which may not reflect the reality. Alternative terms such as "COLMAP-free" or "unknown-pose" might be more precise and aligned with existing literature.

Overall, I believe the strengths of this paper outweigh its weaknesses, and I am therefore glad to increase my rating to Borderline Accept (BA).

**Limitations:**

Yes. The paper mentioned the high computational demands may contribute to higher carbon emission.

**Quality:**

2

**Strengths And Weaknesses:**

**Paper Strengths**

a.	Compared to previous methods that jointly optimize poses and NeRF/3DGS, the paper shows that adding local depth/pose estimation/optimization could lead to better rendering and pose refinement results

b.	The ablation studies are reasonably clear and show effectiveness for each component.

c.	The paper is clearly written and easy to follow, including well-presented figures and tables. I think this is worth giving credit for.

**Weaknesses**

I appreciate the authors’ effort to tackle an important research area, which is the joint optimization of camera poses and 3D scene representation learning. However, I have several concerns that I believe limit the impact of the current submission.

a.	Despite in the relative work section, the paper states that existing novel view synthesis methods still rely on pre-computed COLMAP poses; it does not convincingly demonstrate that NopeRoomGS overcomes these limitations in practice. The proposed method is evaluated on Replica, ScanNet, and selected scenes from Tanks and Temples (please correct me if I’m wrong), which are datasets previously shown to be at least partially solvable by prior works such as BARF, Nope-NeRF, and CF-3DGS. To more rigorously support the claim of being a valid SfM/Pose-free alternative, I believe it should also be evaluated on sequences where COLMAP performs reliably. For example, why not test and report results on All Tanks and Temples scenarios, rather than selecting “advantageous” scenes that “work” like in Nope-NeRF? I think the comparison could be more thorough and show the full picture of the existing methods.

b.	I think the scope of the paper, which mainly focuses on challenging textureless indoor scenes, is narrow compared to real-life use cases. It is unclear if the system is just designed and tuned for indoor pose-free reconstruction. In addition, the proposed local-to-global optimization, seems to be very similar to GS-based SLAM methods, (to name a few: [1], [2], [3], etc.), which also estimate camera poses jointly with 3D Gaussian representations, and many are faster than CF-3DGS or Nope-NeRF. Adding comparisons (for example, on Scannet/Replica) and discussions of these methods would help position NopeRoomGS more clearly within the existing literature. It would be useful to articulate what capabilities NopeRoomGS provides that GS-based SLAM methods do not.

c.	The idea of alternating optimization strategy, while proven to be effective in experiments, isn’t entirely new. Existing feature-based SfM methods like [4] also deploy this type of optimization strategy. However, it would be great if the paper could show a deeper analysis of why alternating optimization is particularly suited to the proposed setting.

d.	I find the ScanNet evaluation lacks important experimental details. It wasn’t clear which scenes were tested on ScanNet. A more detailed description of the experiment setting could improve transparency and reproducibility.

e.	The proposed local-to-global approach adds algorithmic complexity, but the paper does not provide any runtime or memory usage analysis. It would be helpful to include those for better clarity.

f.      Another relevant method to this work is [5], which seems to have better results in Tanks & Temples.

[1] Li24 GS3LAM: Gaussian Semantic Splatting SLAM, ACMMM

[2] Li24 SGS-SLAM: Semantic Gaussian Splatting For Neural Dense SLAM, ECCV

[3] Yan24 GS-SLAM: Dense Visual SLAM with 3D Gaussian Splatting, CVPR

[4] Brachmann25 Scene Coordinate Reconstruction: Posing of Image Collections via Incremental Learning of a Relocalizer, ECCV

[5] Ji25 SfM-Free 3D Gaussian Splatting via Hierarchical Training, CVPR

**Minor Weaknesses**

In Line 326, I’m not sure local neural geometric representation (LNGR) is a good name. How does the representation differ from depth? In addition, it is not introduced or explained in the method section.
In L127-129, “However, existing pose-free methods typically assume small pose perturbation … making them susceptible to failure under large camera motion or in textureless regions.” This claim here may not fully reflect the current state of the field, given that methods like Splatt3r [5] and InstantSplat [6] can handle relatively large camera motion differences with sparse input. I think this is worth double-checking.

[5] Smart24 Splatt3r: Zero-shot gaussian splatting from uncalibrated image pairs

[6] Fan24 InstantSplat: Sparse-view Gaussian Splatting in Seconds

---

> ### Author Rebuttal · Authors · 2025-07-31
>
> # **Reviewer Hqtq**
>
> Dear Reviewer Hqtq,
>
> We sincerely appreciate the valuable feedback.
>
> We address each point below and will further elaborate during the rolling review, given the word limit.
>
> **Q-1: The paper’s claim of overcoming reliance on COLMAP poses is unconvincing. Evaluation is limited to datasets partially solvable by prior methods and omits the scenes in Tanks and Temples, where COLMAP excels. A broader evaluation is needed to support the pose-free claim.**
>
> **A-1:** Thank you for raising this concern. We would like to clarify the misunderstanding regarding our method’s pose-free claim and its evaluation scope.
>
> 1. **Pose-Free Pipeline**
>
>     Our method is indeed pose-free, which does not require external camera poses from COLMAP, SLAM, or IMU. Instead, it jointly optimizes camera poses and 3D Gaussians from monocular RGB inputs via a two-stage process: clip-wise optimization using our Local Neural Geometric Representation (LNGR), followed by global pose refinement and Gaussian merging.
>
> 2. **Evaluation on Tanks and Temples**
>
>     Contrary to the concern, we do evaluate on the full Tanks and Temples dataset, following the standard protocol adopted by CF-3DGS, without cherry-picking. As shown in Table 3 of the manuscript and Table R1-1, our method consistently outperforms CF-3DGS, BARF, and NopeNeRF, even on scenes where COLMAP typically performs well.
>
> 3. **Robustness in Challenging Scenarios**
>
>     As demonstrated in Tables 1–4 and Figure 1 of our manuscript, our method exhibits stronger robustness than prior works (CF-3DGS, NopeNeRF, BARF) across both indoor and outdoor datasets, including Replica, ScanNet, and Tanks and Temples, particularly in textureless scenes and under abrupt camera motion, where many existing methods struggle or fail.
>
>
> We will revise the manuscript to emphasize these points more clearly. Thank you again for highlighting this issue.
>
> *Table R1-1: Camera pose estimation and novel view synthesis performance metrics comparison on Tanks and Temples dataset(mean values).*
>
> | Methods | ATE ↓ | PSNR ↑ | SSIM ↑ | LPIPS ↓ |
> | --- | --- | --- | --- | --- |
> | COLMAP+3DGS | - | 30.20 | 0.92 | 0.10 |
> | BARF | 0.078 | 23.42 | 0.61 | 0.54 |
> | NopeNeRF | 0.006 | 26.34 | 0.74 | 0.39 |
> | NoPoSplat | 0.117 | 20.15 | 0.53 | 0.47 |
> | CF-3DGS | 0.004 | 31.28 | 0.93 | 0.09 |
> | Ours | 0.003 | 31.68 | 0.94 | 0.07 |
>
> **Q-2**:  **The scope is too narrow; The proposed local-to-global optimization seems to be very similar to GS-based SLAM methods. Articulate the difference.**
>
> **A-2:** We thank the reviewer for raising this valuable point.
>
> We provide a detailed clarification below:
>
> 1. **Scope and Generalization.**
>
>     Our method, though designed for challenging indoor scenes, generalizes well to outdoor scenarios without tuning, as shown by strong results on the Tanks and Temples dataset (Sec. 4.4 in the manuscript), which contains outdoor scenes, demonstrating that our method generalizes well beyond indoor cases.
>
>     Moreover, challenging indoor scenarios with textureless environments or abrupt camera motion (e.g., the blank walls or the abrupt movement of the AR kit), targeted by our method NopeRoomGS, are not rare. They are quite common in real-world applications, e.g., AR/VR and indoor assistive robotics.
>
> 2. **Distinctions from GS-based SLAM Methods.**
>
>     (1). **The novelty of our Local-to-global optimization.**
>
>     We acknowledge that the conventional GS-based SLAM methods separate the optimization into a tracking phase and a global optimization phase. Conceptually, our local stage corresponds to the tracking phase in the GS-based SLAM methods (e.g., SGS-SLAM). However, our local stage is fundamentally different from these methods. The conventional methods track the camera frame by frame, which is shown to be unreliable in our setting (Fig.6 in the manuscript).
>
>     Our local stage uses LNGR to jointly process a set of adjacent frames in one batch, offering more robust optimization than frame-by-frame tracking, especially under low texture or abrupt motion.
>
>     (2). **Different assumptions and application domains.**
>
>     GS-based SLAM methods assume high frame rates and small inter-frame motion, relying on temporal continuity by initializing new poses from the previous frame.
>
>     In contrast, pose-free GS reconstruction methods are tailored for offline novel view synthesis and can operate on sequential video input or unordered image collections.
>
>     Our NopeRoomGS requires no temporal ordering for pose initialization and is built to handle abrupt motion and textureless regions where SLAM methods often fail (Table R1-2).
>
> 3. **Comparison with more GS-based SLAM methods.**
>
>     To demonstrate the competitiveness of our method, we compare it with more recent GS-SLAM methods in Table 2 in our manuscript.
>     The method mentioned by the reviewer, *GS-SLAM: Dense Visual SLAM with 3D Gaussian Splatting, has not released its code to date*. And we substitute it with *MonoGS: Gaussian Splatting SLAM*. Following the same protocol and using RGB images as input, the results are reported below.
>
>     *Table R1-2: Comparison of methods in terms of camera pose accuracy and rendering quality on the Replica dataset.*
>
>     | Method | ATE ↓ | PSNR ↑ | SSIM ↑ | LPIPS ↓ |
>     | --- | --- | --- | --- | --- |
>     | GS3LAM | 0.043 | 24.35 | 0.69 | 0.36 |
>     | SGS-SLAM | 0.037 | 25.45 | 0.70 | 0.33 |
>     | MonoGS | 0.039 | 27.40 | 0.73 | 0.29 |
>     | Ours | 0.009 | 32.14 | 0.90 | 0.09 |
>
> **Q-3: Alternating optimization is not new—SfM methods like [4] also use it. Could the paper better explain why it is especially effective in your pose-free 3DGS setting?**
>
> **A-3**: Our alternating optimization is inspired by SfM methods like [4].
>
> However, unlike [4], which optimizes sparse 3D points via geometric reprojection, our pose-free 3D Gaussian Splatting (3DGS) jointly optimizes a high-dimensional set of tightly coupled parameters, including geometry (position, scale, orientation) and appearance (color, opacity), that interact with poses through differentiable rasterization.
>
> This creates a complex, nonlinear photometric loss where errors stem from pose drift, shape misalignment, or appearance mismatch. Under this complex situation, the optimization of camera poses and rendering fidelity could induce gradient conflicts and compromise each other.
>
> The proposed alternating optimization strategy in the global optimization stage can mitigate this issue, resulting in more robust pose estimation and superior rendering quality (See Fig. 6 and Table 5 in our manuscript). Due to the time limit during the rebuttal phase, we will include more analysis on alternating optimization in our final version.
>
> **Q-4: Clarify the selected scene of ScanNet.**
>
> **A-4:** For scene selection, we follow the work, *Neural 3D Scene Reconstruction with the Manhattan-world Assumption* , and use the sequences: scene0616_00, scene0580_00, scene0084_00, and scene0050_00.  We will include these details in the final version and release the data.
>
> **Q-5:  The proposed local-to-global adds complexity, so provide runtime and memory.**
>
> **A-5:** As noted in Sec. 3.1, in the local stage, our LNGR module is lightweight (~30K trainable params), requiring only 2 GB memory,  2 minutes, and 2000 steps per 20-frame clip. For a 100-frame sequence (5 clips), LNGR adds ~10 minutes total overhead.
>
> Our full pipeline runs in <90 minutes per scene on an RTX 4090 with <12 GB peak memory. We will include this runtime analysis in the final version.
>
> *Table R1-4: Comparison of methods in terms of overhead*
>
> | Method | Time(min) | Memory(GB) |
> | --- | --- | --- |
> | NopeNeRF | ~400 | 6 |
> | CF-3DGS | ~63 | 9 |
> | Ours | ~81 | 12 |
>
> **Q-6:** **Comparison with  SFGS-HT, which seems to perform better in the Tanks dataset.**
>
> **A-6:** We thank the reviewer for highlighting this relevant work. We acknowledge that the concurrent work, SFGS-HT: *SfM-Free 3D Gaussian Splatting via Hierarchical Training*, performs better on the Tanks and Temples dataset, but may fail to converge on indoor scenes with severe motion and low-texture regions. We evaluate SFGS-HT on the indoor dataset Replica. Its performance degrades easily, as shown in Table R1-5, while our method shows robustness.
>
> *Table R1-5: Comparison of methods in terms of camera pose metrics and rendering quality on the Replica dataset.*
>
> | Method | ATE ↓ | PSNR ↑ | SSIM ↑ | LPIPS ↓ |
> | --- | --- | --- | --- | --- |
> | SFGS-HT | 0.021 | 26.11 | 0.78 | 0.18 |
> | Ours | **0.009** | **32.14** | **0.90** | **0.09** |
>
> **Q-7: Minor Weaknesses: (1) LNGR is not a good name. How does the representation differ from depth? (2) Clarify the difference with pose-free methods, such as Splatt3R and InstantSplat.**
>
> **A-7:** Thank you for your comments.
>
> (1) LNGR builds on a pretrained CNN depth model to jointly predict depth over short clips. It is finetuned with multi-frame geometric consistency to correct scale drift from monocular depth estimation (see Sec. 3.1 of our paper). Conceptually, it learns a consistent local geometric representation. We will revise the module name for clarity.
>
> (2) Recent feed-forward methods like Splatt3r and InstantSplat directly output 3DGS representation, taking several input views,(e.g., 2 frames in Splatt3r, ~10 frames in InstantSplat) , recovering limited scene. These types of methods focus on the generalization, usually have difficulties in high-resolution rendering  (only output 512x512) due to their high memory consumption. In contrast, our method follows the track of a per-scene optimization pipeline (similar to SfM, NeRF, 3DGS), focusing on how to robustly optimize the representation for each scene separately.

---

> ### Author Response · Authors · 2025-08-01
>
> Dear Reviewer Hqtq,
>
> Thank you very much for your thoughtful and detailed review. We truly appreciate your time and constructive feedback. We have carefully addressed your concerns in our rebuttal, including:
> - **Clarifying our evaluation scope on Tanks and Temples**, where we follow the standard protocol and report performance across all scenes—achieving superior results even where COLMAP works well.
> - **Clarifying the Distinctions from the GS-based SLAM approaches**, which highlights our robustness in textureless or fast-motion indoor scenes, and offers head-to-head comparisons on Replica.
> - **Explaining the distinct challenges** of optimizing high-dimensional 3DGS parameters in a pose-free setting, where our alternating optimization strategy proves especially effective.
> - **Providing runtime and memory analysis**, confirming our method's efficiency compared to baselines like CF-3DGS and NopeNeRF.
>
> We believe these updates more clearly support our claims and better situate our contributions within the broader literature.
> We hope these clarifications may help alleviate your concerns. If you find our responses satisfactory, we would sincerely appreciate your consideration of a higher score. We look forward to your further input during the discussion phase. Thank you again for your valuable feedback and engagement.
>
>
> Warm regards,
> Submission2437 Authors

---

> > ### Comment · Reviewer_Hqtq · 2025-08-02
> >
> > I appreciate the authors’ rebuttal and their efforts in addressing my initial concerns. I hope the authors don’t mind if I raise a few follow-up questions for further clarification.
> >
> > **A-1:**
> > Could the authors kindly confirm whether the experiments reported in Table 3 of the main paper/Table R1-1 of the rebuttal were conducted on all 17 scenes of the Tanks and Temples dataset? For example, Figure 7 of [4] presents results on the complete dataset, while Nope-NeRF (Table 1) and CF-3DGS (Table 2) report results on only 8 of the 17 scenes.
> >
> > It is my understanding that Nope-NeRF and CF-3DGS may not yet generalize to all scenes in this dataset, which introduces a certain degree of selection bias. Table R1-1 in the rebuttal also appears to use the 8-scene performance as in CF-3DGS (table 2), which could lead to some confusion. I would appreciate clarification on this point.
> >
> > **A-2:**
> > Could the authors also clarify the source of the discrepancy between the GS3LAM [1] results shown in Table R1-2 and those reported in Table 2 of the original GS3LAM paper [1]? Are the differences attributable to changes in experimental settings or evaluation protocols?
> >
> > **A-3:**
> > Thank you for the explanation regarding the optimization strategy. Since the proposed alternating optimization approach draws inspiration from SfM, I would suggest briefly acknowledging this connection in the related work section. Doing so would not detract from the novelty of the contribution, but rather help position the method more clearly in relation to existing literature.

---

> ### Author Response · Authors · 2025-08-03
>
> **Q-1: Could the authors kindly confirm whether the experiments reported in Table 3 of the main paper/Table R1-1 of the rebuttal were conducted on all 17 scenes of the Tanks and Temples dataset? It is my understanding that Nope-NeRF and CF-3DGS may not yet generalize to all scenes in this dataset, which introduces a certain degree of selection bias.**
>
> **A-1**: Thank you for your follow-up question. We now realize we may have misunderstood your original concern in Q1. We initially thought you suspected that we had intentionally selected a subset from the 8 scenes of Tanks and Temples (adopted by the prior works) for evaluation. However, it seems your concern is that the 8-scene selection, adopted by prior works like Nope-NeRF, CF-3DGS, FreGS, Flow Nerf, TD-NeRF, etc, may already introduce bias. We sincerely apologize for the misinterpretation.
>
> To clarify: we followed the same 8-scene evaluation protocol on the Tanks and Temples dataset used in previous pose-free works, Nope-NeRF and CF-3DGS, to ensure a fair comparison. We understand your concerns about the scene selection bias caused by the prior works. We agree that evaluating all 17 scenes would better validate our method’s generalizability. As we cannot provide additional results during this phase, we will try to extend our experiments to include the additional results on the Tanks and Temple in the final version of the paper.
>
> However, we also want to emphasize again that **our method mainly focuses on improving the 3DGS in textureless indoor environments and abrupt camera motion**. So the **indoor datasets, Replica and ScanNet**, are our main focus (the results are summarized in Tables 1 and 2).  The evaluation on the Tanks and Temples dataset is only to demonstrate its capacity to generalize beyond its intended domain, as stated in **L102-L104** and **L267-L269** in our manuscript.
>
>
> **Q-2: Could authors clarify why GS3LAM results in Table R1-2 differ from those in its original paper, different evaluation protocols?**
>
> **A-2**: Thank you for pointing this out. GS3LAM is an **RGB-D-based** method that leverages **GT depth** for both camera pose estimation and reconstruction. In contrast, our method is designed to operate under a more **practical and realistic RGB-only setting**, where **depth sensors are not available**, a common scenario in real-world applications such as consumer-grade mobile devices or Internet-scale photo collections.
>
> For a fair comparison,  we restricted all methods to RGB input. Specifically, we re-evaluated GS3LAM using the same predicted depth from our foundation model, replacing GT depth. This aligns both methods under the RGB-only assumption.
>
> The results in Table R1-2 reflect this RGB-only protocol. While they differ from GS3LAM’s original Table 2, we believe this setting better assesses real-world applicability and the challenges our method aims to address.
>
> **Q-3: Since the proposed alternating optimization approach draws inspiration from SfM, I would suggest briefly acknowledging this connection in the related work section.**
>
> A-3: We appreciate the suggestion and fully agree. While our alternating optimization strategy addresses new challenges specific to dense differentiable rendering with 3DGS, its roots in classical SfM methods (e.g., [4]) are clear and worth acknowledging.
>
> We will revise the related work section to explicitly discuss this connection. This will better situate our contribution in the broader context of optimization strategies for pose and scene estimation.

---

> > ### Author Response · Authors · 2025-08-06
> >
> > Dear Reviewer Hqtq,
> >
> > We sincerely appreciate your thoughtful follow-up questions and suggestions. We’ve provided responses to each of your points above, and we hope they offer the necessary clarifications.
> >
> > If there are any remaining concerns, we would be more than happy to respond. Your feedback is valuable to us, and we deeply appreciate the time and effort you've taken in reviewing our work.
> >
> > Warm regards,
> >
> > Submission2437 Authors

---

> > > ### Comment · Reviewer_Hqtq · 2025-08-08
> > >
> > > Thank you to the authors for the detailed rebuttal. I believe the majority of questions have been clearly addressed.
> > >
> > > At this point, I have no major remaining concerns with the paper. However, I would like to offer two comments to conclude my response:
> > >
> > > 1. The authors clarified that only eight scenes from the Tanks and Temples dataset were evaluated, which is the same protocol being used in some of the previous baselines. This detail (which specific scenes) should be explicitly stated in the final paper to avoid potential confusion. Since the authors have promised to include full dataset results in the camera-ready version, I am leaning towards raising my rating.
> > >
> > > 2. I also agree with Reviewer Ht6p that the term "pose-free" may feel somewhat strong or idealized, given the current state of the field. While not technically incorrect, it could give readers the impression that the problem is well solved, which may not reflect the reality. Alternative terms such as "COLMAP-free" or "unknown-pose" might be more precise and aligned with existing literature.
> > >
> > > Overall, I believe the strengths of this paper outweigh its weaknesses, and I am therefore increasing my rating to Borderline Accept (BA).

---

> > > > ### Author Response · Authors · 2025-08-08
> > > > **Thank you for your positive final rating**
> > > >
> > > > Dear Reviewer Hqtq,
> > > >
> > > > Thank you for your positive feedback and for providing these valuable final comments.
> > > >
> > > > We will explicitly state the evaluation protocol of Tanks and Temples dataset used in our experiments in the final paper to avoid any ambiguity, and include the full-dataset results as promised. Regarding terminology, we appreciate your perspective on the use of “pose-free” and will carefully consider alternatives such as “COLMAP-free” or “unknown-pose” to better align with existing literature and avoid potential misinterpretation.
> > > >
> > > > We sincerely appreciate your thoughtful review and the time you have dedicated to our submission.
> > > >
> > > > Sincerely,
> > > >
> > > > Submission 2437 Authors

---

> > ### Comment · Area_Chair_nUhV · 2025-08-08
> > **detailed response**
> >
> > Dear Reviewer Hqtq,
> >
> > Please provide your detailed response to the rebuttal. Thanks!
> >
> > Best regards, AC

---

### Note · Authors · 2025-08-12

Dear Area Chair and Esteemed Reviewers,

Thank you for your thoughtful comments and valuable suggestions.

We are truly grateful for the positive final ratings from all reviewers.

We appreciate the recognition of our key contributions, particularly (1) enabling pose-free 3D Gaussian Splatting reconstruction in indoor environments with textureless regions and abrupt camera motion, and (2) demonstrating robust performance across the Replica, ScanNet, and Tanks and Temples datasets, confirming our method’s effectiveness in overcoming the limitations of existing frameworks in challenging scenarios.

Following the reviewers' guidance, we further clarified the distinctions between GS-based reconstruction and GS-based SLAM methods, and provided an analysis of our framework's efficient runtime and memory consumption. We are excited to know that these clarifications have resolved the reviewers' concerns. The additional discussions from our rebuttal will be integrated into the final version of the manuscript.

Thank you again for your time and valuable suggestions.

Best regards,

The Authors of Submission 2437

---

### Decision · Program_Chairs · 2025-09-17

**Decision:**

Accept (poster)

**Comment:**

This paper introduced Pose-Free Indoor 3D Gaussian Splatting technique. Initially, the reviewers raised concerns regarding comparisons to GS-based SLAM methods, memory and time comparisons with previous methods, and comparisons with methods such as VGG-SfM, VGG-T, or DROID-SLAM. However, after the rebuttal, the authors successfully addressed most of these concerns, including the clarification of novelty. The AC has also reviewed the paper, reviewer comments, and rebuttal, agreeing that the paper is well-motivated, clearly written, and supported by thorough experiments. Therefore, the AC recommends acceptance. It would be beneficial to incorporate all additional experiments and discussions from the rebuttal into the final version.